

# Prior heterogeneous ice nucleation events increase likelihood of homogeneous freezing during the evolution of synoptic cirrus

Kasper Juurikkala [1,2],  Christina J. Williamson[1,3],  Karl D. Froyd[4,5],  Jonathan Dean-Day[6], and Ari Laaksonen[1,7]

[1]Finnish Meteorological Institute, 00560 Helsinki, Finland
[2]Department of Physics, University of Helsinki, Helsinki, 00014, Finland
[3]Institute for Atmospheric and Earth System Research/Physics, University of Helsinki, Helsinki, 00014, Finland
[4]Cooperative Institute for Research in Environmental Sciences, University of Colorado, Boulder, Colorado 80309, USA
[5]Air Innova Research and Consulting, Boulder, CO 80305, USA
[6]Bay Area Environmental Research Insititute, Moffett Field, CA 94035, USA
[7]Department of Technical Physics, University of Eastern Finland, 70211 Kuopio, Finland

**Correspondence:** Kasper Juurikkala  (kasper.juurikkala@fmi.fi)

**Abstract.** In-situ observations are currently used to classify synoptic cirrus as formed by homogeneous or heterogeneous ice nucleation based on ice residual analysis. We use large-eddy model UCLALES-SALSA to show the limitations of this method by demonstrating that prior heterogeneous ice nucleation events can shape the thermodynamic conditions for homogeneous freezing to occur more likely in the subsequent nucleation events.

In a case study of synoptic cirrus from NASA's Midlatitude Airborne Cirrus Properties Experiment (MACPEX), observations suggest homogeneous freezing as the dominant nucleation mechanism. Simulations done with UCLALES-SALSA show that homogeneous freezing occurred after earlier heterogeneous ice nucleation events, with mineral dust acting as the ice-nucleating particles (INPs). Heterogeneous ice nucleation depleted INPs from cirrus forming altitudes, creating favourable conditions for homogeneous freezing at the time of observations.

This study modelled cirrus cloud properties based on measured conditions and compared simulated results with observed cloud structures. It is shown that modelling the impact of prior nucleation events on the vertical distribution of mineral dust and humidity in the model is necessary to reproduce the observed cloud characteristics. Heterogeneous ice nucleation primarily had a role in removal of ice-nucleation active mineral dust from cloud-forming altitudes well before arriving at the measurement location, while having limited role in forming ice crystals shortly before the time of measurements.

Model results also show that small-scale wave activity strongly influenced ice nucleation efficiency and overall cloud properties. While large-scale atmospheric dynamics typically dominate synoptic cirrus formation, they alone were insufficient to replicate the observed cloud characteristics.

## 1 Introduction

The 2021 IPCC report (Forster et al., 2021) highlights that clouds and aerosols remain one of the primary sources of uncertainty

in the Earth's energy budget and our ability to predict future climate. The intricate interactions between these atmospheric com-





ponents pose significant challenges for climate modelling, particularly because microphysical processes occur at small spatial scales that cannot be directly resolved in general circulation models (GCMs) (Burrows et al., 2022). As a result, parametrizations are necessary, requiring a delicate balance between simplicity, realism, computational stability, and efficiency (Boucher et al., 2013).

Another substantial source of uncertainty in cloud radiative effects arises from the large knowledge gaps regarding humidity, particularly ice supersaturated regions in the upper troposphere and lower stratosphere (UTLS), where in-situ cirrus clouds form. At present, information on the large-scale transport of humidity in the UTLS primarily comes from numerical weather prediction (NWP) models, such as the Integrated Forecasting System (IFS) developed by the European Centre for Medium-Range Weather Forecasts (ECMWF, 2016) and the ICOsahedral Non-hydrostatic model (ICON; Zängl et al., 2015; Seifert and

Siewert, 2024), used by the German Weather Service. These NWP models rely heavily on observational data to maintain a realistic representation of atmospheric conditions. However, the scarcity of spatially and temporally resolved observations in the UTLS hinders their ability to accurately predict ice supersaturation ($S_i$).

Over the past few decades, several key measurement campaigns (e.g., Krämer et al., 2009; Voigt et al., 2017) have been conducted in the UTLS. These campaigns have shown that high $S_i$ is more common than predicted by NWPs. Such high $S_i$ levels

promote the formation of UTLS cirrus clouds, which form synoptically in the absence of convection. However, given the uncertainties surrounding $S_i$, the formation mechanisms and occurrence of these synoptically driven cirrus clouds remain poorly quantified.

Synoptic cirrus clouds primarily form through two dominant mechanisms: heterogeneous and homogeneous freezing. Heterogeneous ice nucleation occurs either via immersion freezing, where insoluble particles—commonly referred to as ice-

nucleating particles (INPs)—are embedded in aqueous droplets and trigger freezing, or via deposition nucleation, where water vapour directly deposits onto dry, insoluble particles and freezes. For immersion freezing to occur, the INPs have to be somewhat hydrophilic or acquire a water-soluble coating during atmospheric transport. Electron microscopy studies (e.g., Kojima et al., 2006; Cziczo et al., 2013; Twohy, 2014) of cirrus crystal residuals have found that a substantial fraction of these particles consist of uncoated mineral dust. As coatings on mineral dust tend to suppress their heterogeneous ice nucleation efficiency

(shifting the nucleation threshold toward higher supersaturation), deposition nucleation is likely the prevailing formation mechanism in many cases. In contrast, homogeneous freezing occurs in the absence of INPs and takes place when aqueous solution droplets freeze at temperatures below the -38°C threshold for pure water and at high $S_i$.

Ice nucleation in cirrus clouds is strongly influenced by the abundance of INPs, which regulate how efficiently heterogeneous ice nucleation can suppress homogeneous freezing and activity. Heterogeneous ice nucleation activates and removes a popu-

lation of ice-nucleation-active INPs from altitudes where they originate and end up lower in altitude through sedimentation of ice crystals. Cirrus clouds formed predominantly through heterogeneous ice nucleation tend to have ice crystal concentrations closely tied to the availability of INPs. In contrast, cirrus formed via homogeneous freezing often exhibit a much wider range of ice crystal concentrations, depending on environmental factors such as temperature and vertical velocity (Kärcher and Lohmann, 2002).

Currently, the determination of whether a cirrus cloud formed through heterogeneous or homogeneous freezing is typically



inferred from ice residual analysis. However, while this method provides valuable insights, it does not capture the processes leading to the observed state of the cirrus cloud. The objective of this study is to use the UCLALES-SALSA large-eddy model (Tonttila et al., 2017) to illustrate the complexity of cirrus cloud formation that cannot be fully explained by ice residual analysis alone. This work focuses on a case of cirrus cloud case observed during the Midlatitude Cirrus Properties Experiment

MACPEX campaign (Jensen et al., 2013b), which predominantly exhibited homogeneous freezing among the analysed ice crystals (Cziczo et al., 2013). This raises a key question: why did this particular cirrus cloud form primarily through homogeneous freezing?

## 2 Methods

### 2.1 MACPEX campaign

The MACPEX campaign conducted by NASA was an aircraft measurement campaign aimed at investigating cirrus cloud properties (Jensen et al., 2013b). The campaign took place between March and April 2011, involving multiple science flights with the NASA WB-57F science aircraft over the southern United States. A total of 14 science flights were conducted, focusing on synoptic and anvil cirrus clouds, resulting in over 18.5 hours of cirrus sampling. In addition to in-situ measurements, remote-sensing observations were coordinated to target cirrus clouds along the flight paths.

### 2.2 Description of the instruments used in the analysis

Ice number concentration, size distribution and IWC was measured by several instruments. The two-dimensional stereo (2D-S; Lawson et al., 2006) probe is an optical imaging system which uses two pairs of probes to shoot orthogonal laser beams to 128-photodiode arrays. Passing objects create a shadow which is observed on the sensors (Lawson et al., 2006). The 2D-S is capable of detecting particles with sizes from 10 $\mu$m to over 1 mm in diameter which is a good range for capturing the evolu-

tion of the crystal sizes in ice clouds. The 2D-S determines the crystal size and habit by using algorithm described in Lawson (2011) and results in filtration of biased data created by shattering crystals. The data for the very first size bin (5-15 $\mu$m) of the 2D-S instrument is excluded in the analysis due to known over-estimations of the number concentrations, and this has been addressed in multiple studies (e.g. Jensen et al., 2013b; Krämer et al., 2016). The limited reliable observation capability of the 2D-S probe above 15 $\mu$m restricts obtaining accurate information about young cirrus clouds with high number concentrations

of smaller-sized particles (Krämer et al., 2016).

Meteorological variables such as temperature, pressure, wind direction and speed, geographic location, and speed of the aircraft were measured with the Meteorological Measurement System (MMS; Scott et al., 1990). The MMS vertical wind component measurements are filtered by subtracting 20 or 150 second (approximately corresponding to the model domain width and mesoscale dynamics) local mean trend of wind from the 20 Hz measurements. Additionally vertical wind measurements from

constant altitude legs are limited to a domain constrained by aircraft vertical velocity values below +/- 1 m s$^{-1}$ in magnitude. The humidity was measured with multiple instruments, with the Harvard water vapour (HWV) instrument being employed in



this study.

The NOAA Particle Analysis by Laser Mass Spectrometry (PALMS; Thomson et al., 2000; Froyd et al., 2019) instrument measures the size-resolved chemical composition of particles in real time within the size range of 0.15 to 5.0 $\mu$m (Thomson

et al., 2000; Cziczo et al., 2013). The instrument operates by directing sample air into a vacuum system where an excimer laser ablates and ionizes the particles. The ionized particles are then analysed with a time of flight mass spectrometer (Murphy et al., 2006), and through post-processing, the particles can be classified based on their composition (Froyd et al., 2009). To obtain particle type-resolved number size distributions, the PALMS measurements were combined with those obtained from a focused cavity aerosol spectrometer (FCAS II; Jonsson et al., 1995). The FCAS II measures the size resolved particle number

concentration within a measurable size range of 0.07 to 1.5 $\mu$m. Initially, the particles pass through an anisokinetic sampler, which slows them down, and then they are transported to a laser cavity. Inside the cavity, the particles traverse a laser beam, and the scattering light is measured to determine their size (Jonsson et al., 1995).

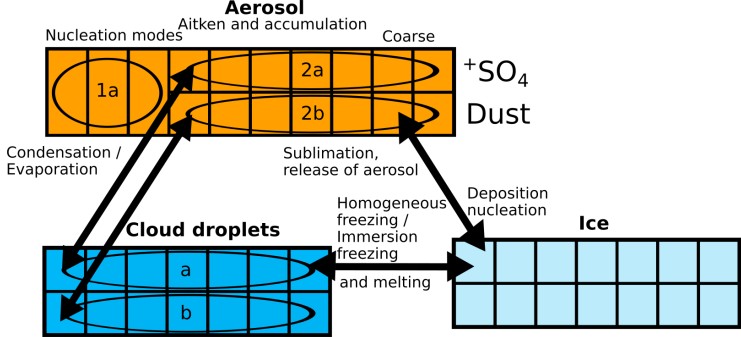

**Figure 1.** Bin scheme used in UCLALES-SALSA to track particle types relevant to ice nucleation in synoptic cirrus clouds. Freezing of rain is also possible while it is not shown here as it generally does not occur within the realms of this study.

## 2.3 UCLALES-SALSA

UCLALES-SALSA (Tonttila et al., 2017) is a large eddy (LES) model with a combination of sectional aerosol bin microphysics model SALSA (Sectional Aerosol module for Large-Scale Applications) (Kokkola et al., 2008; Tonttila et al., 2017; Kokkola et al., 2018). UCLALES is a well known atmospheric LES model based on the work (Stevens et al., 1999, 2005). The SALSA module enables tracking of four types of particles (including aerosols, cloud droplets, rain/drizzle particles and ice), and their particle size and chemical composition in a bin scheme, shown in Fig. 1. The aerosols are categorized into two

distinct bins: one range, from 1a to 2a, addresses particles ranging from nucleation to coarse modes (3 nm - 10 $\mu$m), while the other, bin 2b, encompasses sizes above the Aitken and accumulation mode (50 nm - 10 $\mu$m). Particles falling within subrange 1a (3 - 50 nm) typically originate from new particle formation (e.g., sulfates) and may include some primary organic particles (Kokkola et al., 2008). This binning strategy enables separate tracking of two aerosol populations with an example: 2a for sol-



uble sulfates crucial in cloud activation, and 2b for mineral dust particles relevant to ice nucleation. Furthermore, this parallel
tracking extends to cloud and ice bins, facilitating comparison between the evolution of these two aerosol populations.

SALSA supports both heterogeneous and homogeneous freezing mechanisms. Of known heterogenous freezing mechanisms, immersion, contact freezing and deposition nucleation have been implemented. Immersion freezing can occur when an aqueous solution droplet has an insoluble core such as mineral dust or black carbon. Contact freezing does not have a separate implementation in SALSA and is included in the immersion freezing scheme. Deposition nucleation can occur only for dry
insoluble uncoated particles above supersaturated condition over ice ($S_i > 1$), however, below saturation over water.

As illustrated in Fig. 1, once ice-nucleation-active aerosols nucleate ice, they are removed from the aerosol bins and transferred to the ice bins. Within these ice bins, ice crystals can either grow or shrink through vapour deposition or evaporation, and they are sorted into their respective size bins. When the ice crystals evaporate, the INPs are returned to the aerosol bins. The microphysics governing ice crystal vapour growth is based on Jacobson (2005).

## 3   Case study-16 April 2011


A mission flight with the NASA WB-57F was flown on 16th of April to target cirrus cloud trail flowing easterly. The plane took off at 1700 UTC from Houston, Texas, and flew directly towards the cirrus clouds. Before flying into the cirrus clouds, lower stratospheric air was sampled and also provided the meteorological conditions over the cirrus cloud layers. The sampling of cirrus was done by flying back and forth following the cirrus cloud trail. The flight path is presented in Fig. 2.

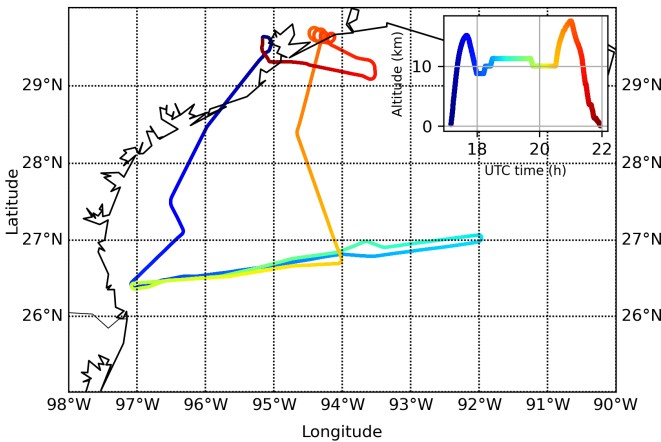

**Figure 2.** Flight path of WB-57F on April 16, 2011 shown with coloured lines. The colour of line on the map correlates the geographical position of the aircraft to the altitude and time in UTC.





### 3.1 Meteorological background

On April 16, 2011, the meteorological conditions over the Southwest Continental United States and Northern Mexico were characterized by the dominance of multiple high-pressure systems. Additionally, a cold front was associated with a low-pressure system in the Eastern United States, and its trailing edge extended over the Gulf of Mexico. Meanwhile, a long trail

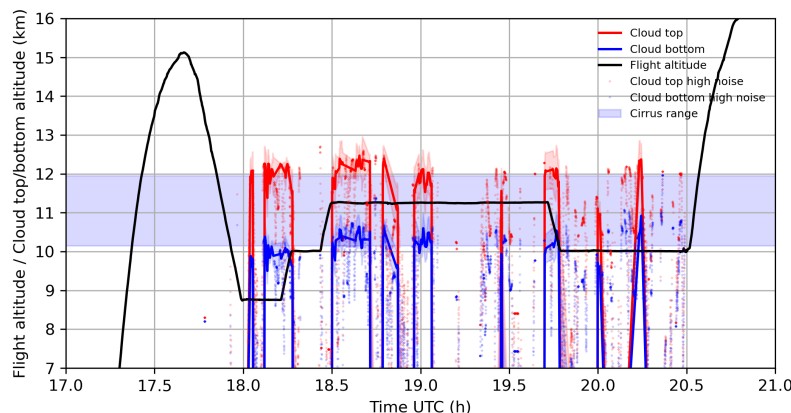

**Figure 3.** Cloud top and bottom altitude derived from GOES-16 measurements along the flight altitude plotted against the time. Filtering was applied to the cloud top and bottom data due to the high uncertainty in data. The data with small uncertainty is connected with lines to illustrate better the approximate cloud top and bottom heights.

of patchy cirrus clouds were forming over the Northern parts of Mexico and flowing easterly towards the Gulf of Mexico. According to ECMWF ERA5 reanalysis data, these cirrus clouds formed in a moisture rich layer (10-12 km) originating from the Pacific Ocean, carried by a subtropical jet.

Figure 3 shows a derived product of cirrus cloud top and bottom altitudes, based on measurements from the Geostationary Operational Environmental Satellite-16 (GOES-16), operated by the National Oceanic and Atmospheric Administration (NOAA),
correlated to the location of the WB-57F aircraft, where 2 km thick cirrus cloud layers between 10 and 12 km along the flight path can be seen. Figure 4 presents the vertical profile of meteorological variables during aircraft ascents and descents in the proximity of cirrus clouds. The vertical layer where the cirrus clouds are present is stable throughout as the potential temperature gradient is significantly above 0 ($\frac{d\theta}{dz} > 0$). This implies that the formation of these cirrus clouds is influenced by large-scale forcing, classifying them as synoptic cirrus clouds. The supersaturation over ice ($S_i$) within the altitudes of cirrus
clouds exhibits notable variability. As depicted in Fig. 3, the ascends and descents do not always intersect cloudy regions, indicating that the humidity data in Fig. 4 represents a mixture of clear air and cloudy air conditions.

Also, the ECMWF ERA5 reanalysis shows a significant spread in values for $S_i$ along the WB-57F flight path. Most of the observed $S_i$ data points fall within the range of values provided by ERA5. However, $S_i$ is almost entirely below 1 in the ERA5 data, indicating that cirrus cloud formation is under-represented in the reanalysis. This under-representation is further





confirmed by the low values of cloud ice water mixing ratio along the flight path in the ERA5 dataset.

The formation mechanism of cirrus clouds is investigated through back-trajectory calculations conducted with Lagrangian analysis python script, utilizing wind field data from ERA5. As illustrated in Fig. 5, air parcels originating at 10 km appear to undergo major uplift around 4-7 UTC, influenced by topography interacting with the prevailing air-mass over the Western parts of Mexico with mountainous terrain. This uplifting is connected to the rapid development of high ice water content cirrus

clouds, as evidenced by satellite imagery in the region as shown in Fig. 6. The presence of mountain ranges induces gravity waves (Smith, 1979; Jensen et al., 1998; Joos et al., 2008, 2009), generating local ascending and descending air patterns downstream from the point of disturbance, even present during the time of WB-57F measurements over the ocean (between 18-20.5 UTC in Fig.5). The data shown strongly suggests that the cirrus cloud formation was induced by these gravity waves with periods of couple hours.

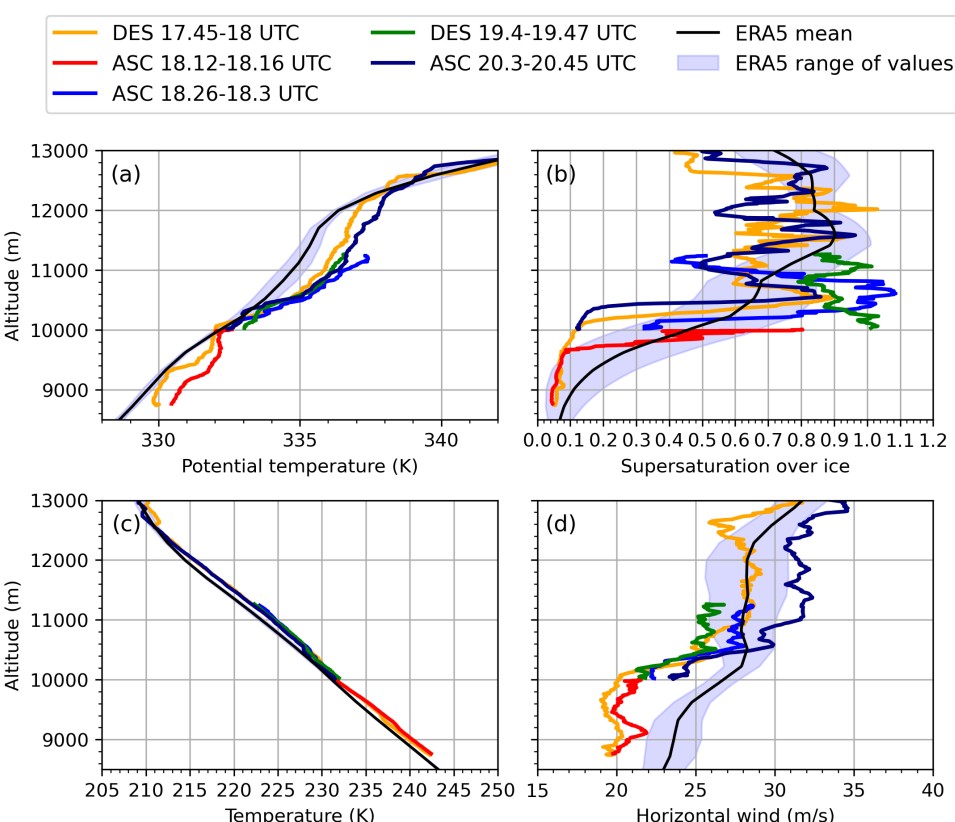

**Figure 4.** Potential temperature (a), temperature (b), supersaturation over ice (c) and horizontal wind (d) measured with MMS during the ascents and descents of the aircraft. The vertical profiles are result of multpiple ascents (ASC) and descent (DES) through the cirrus cloud altutudes. ECMWF ERA5 data is shown with shading of light blue representing the 1-$\sigma$ around the mean of altitude.




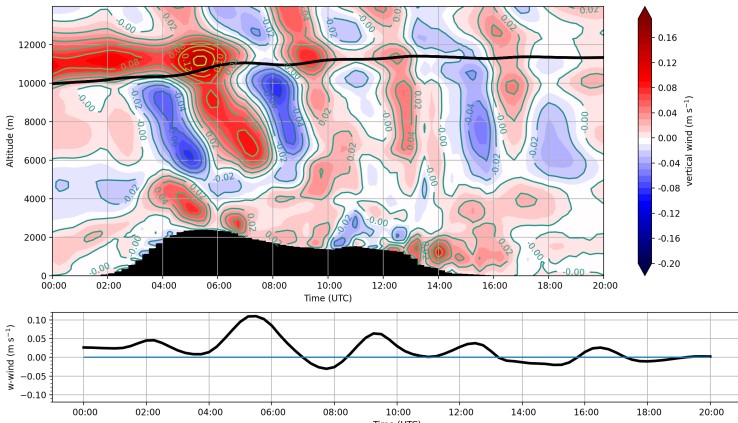

**Figure 5.** Vertical profile of vertical wind along the back-trajectory analysis moving along the cirrus layer. Black line shows the movement of air parcel starting at 10 km at 0 UTC. The surface elevation is in black. The vertical wind around the air-parcel is shown in the lower panel

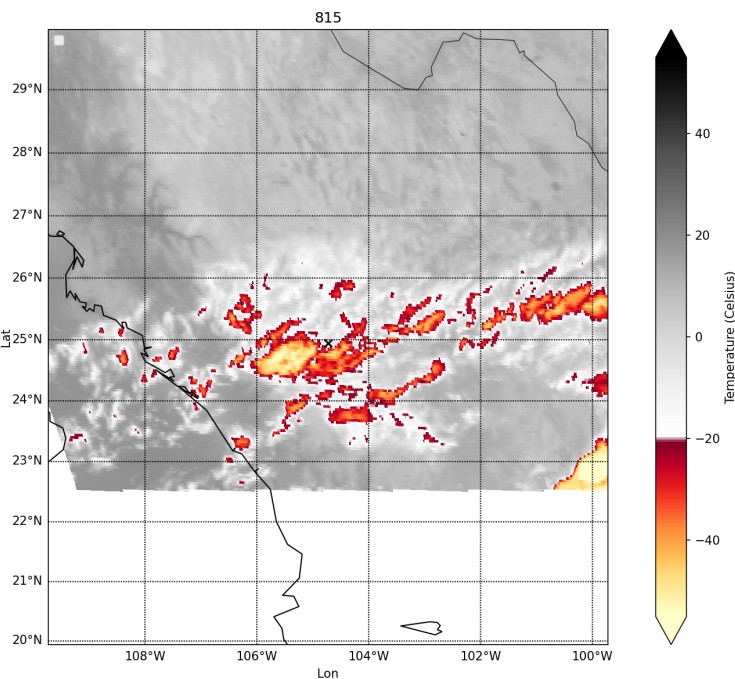

**Figure 6.** Bright cirrus clouds around 8 UTC with temperatures below -40 °C. Temperature corresponds to the effective cloud top temperature of the clouds. Imagery taken with GOES-16 satellite. The center is marked with cross which indicates a location of an airparcel at 8 UTC which is traced to the location of WB-57F measurement later in time.




## 3.2 Indication of homogeneous freezing


The April 16th cirrus case was the only observed instance during the MACPEX campaign where the primary nucleation mechanism was predominantly homogeneous freezing (Cziczo et al., 2013). This conclusion is supported by examining the ice residual particle (IRP) populations. Particles composed of a mix of sulfates, organics, nitrates, and biomass burning particles contain a substantial fraction of water-soluble compounds (Reid et al., 2005), which promote homogeneous nucleation at tem-

peratures below 235 K (Koop et al., 2000). In contrast, high IRP fractions of mineral dust and metallic particles are generally indicative of heterogeneous nucleation (Cziczo et al., 2013). Observations from April 16th revealed substantial fractions of sulfate/organic/nitrate particles and biomass burning particles in IRP, with a much lower fraction of mineral dust compared to heterogeneously nucleated cirrus cases presented by Cziczo et al. (2013) (see their Fig. 6). Interestingly, mineral dust fractions were higher in the observed IRP compared to clear-air aerosol fractions (Fig. 7), suggesting limited influence of mineral dust

on the ice crystal population. This indicates that at least some ice likely formed heterogeneously, albeit not predominantly. Cziczo et al. (2013) noted that mineral dust and metallic particles, when acting as heterogeneous INPs in cirrus clouds, are typically thinly coated, making them effective for deposition nucleation. However, PALMS spectral analysis showed that sulfate coatings on April 16th were thicker than on other MACPEX flight days (analysis results available in the supplementary material). Although the data quality is limited, this suggests that well-aged UTLS air results in thicker coatings on mineral dust

particles, which may suppress their heterogeneous ice nucleation ability, as has been shown in previous studies (e.g., Cziczo et al., 2009; Eastwood et al., 2009; Chernoff and Bertram, 2010; Sullivan et al., 2010).

PALMS measurements also included uncharacterized particles labeled as "other", which did not have obvious characteristics

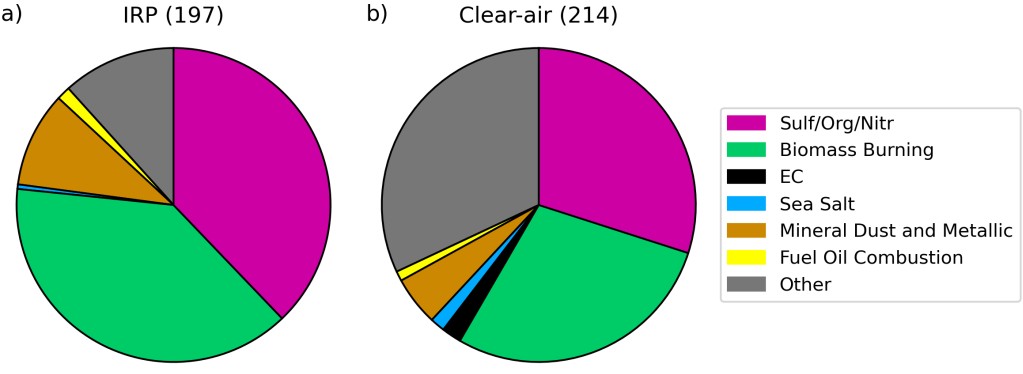

**Figure 7.** Ice cloud residual (a) and clear-air (b) contribution of aerosol species in a range from 0.2 to 3$\mu$m for IRP and 0.2 to 1.5$\mu$m for clear air.

of heterogeneous INP. These accounted for approximately one-third of the clear-air aerosol population, a greater proportion





than observed during other MACPEX flights. These particles were compositionally typical of the region, with carbon-rich
spectra inconsistent with sulfate/organic mixtures, and included some with pyridinium signatures.

There are some limitations to the analysis of IRP and clear-air particle compositions in this study. By definition, the measured
clear-air aerosols are those that did not nucleate into ice at the time of measurement. Alternatively, these aerosols may have
been involved in ice nucleation events and subsequent cloud sublimation well before the measurements were taken. Conse-
quently, it is unclear whether these aerosols exhibit the same ice nucleation properties as those measured inside the cloud
(IRP). Furthermore, the mineral dust particle fractions are averaged from a very limited clear-air sample set, making direct
comparisons with IRP challenging. Additionally, the IRP data is inherently biased toward smaller ice crystals, which could
potentially skew the results toward higher fractions of homogeneous IRP. However, clear-air particle fractions on Apr 16 were
similar to the regional MACPEX averages.

In Fig. 8a, the measured $N_i$ is presented and it exceeds the concentration of mineral dust particles or other potential hetero-
geneous INPs, providing strong evidence that homogeneous freezing played a significant role in shaping the $N_i$ distribution.
Homogeneous freezing typically requires $S_i$ levels within the range of 1.46–1.54 at temperatures between 210 and 230 K (cal-
culated with Eq. 10 in Ren and Mackenzie (2005)). The MMS measured $S_i$ in Fig. 8b reveals a notable absence of $S_i$ values
above 1.2, however, the measurements mostly represent either environment in the absence of clouds or inside matured cirrus
clouds where the high $S_i$ would be difficult to find. Observing high $S_i$ required for homogeneous freezing is inherently chal-
lenging, especially within fully developed cirrus clouds, as the available humidity rapidly decreases following a homogeneous
freezing event. This reduction is driven by nucleated ice crystals and sedimenting ice crystals from higher altitudes, which de-
plete the water vapour within the layer. Numerical studies (e.g., Spichtinger and Cziczo, 2010) and in-situ measurements (e.g.,
Jensen et al., 2013a) consistently show that $S_i$ quickly returns to near-equilibrium levels after homogeneous freezing occurs,
making it unlikely to detect significantly elevated $S_i$ in measurements. These limitations highlight the difficulty of directly
capturing the supersaturation levels required for homogeneous freezing in such environments.

### 3.3 Cloud vertical structure

Figure 8 illustrates the distribution of $N_i$ observed during the mission flight. A shift in the distribution of $N_i$ measured with
2DS between two altitude levels is visible. The median $N_i$ is about an order of magnitude lower in the lower parts of cirrus
which could be explained by following factors:

– Homogeneous and heterogeneous ice nucleation produces higher number of ice when the temperatures are lower (Jensen
  et al., 2013b), leading to higher $N_i$ in the upper parts of cirrus.

– The WB-57F collected statistically significant data between 9–11.2 km, covering only the lower portion of the cirrus
  cloud, where $S_i$ is strongly influenced by sedimenting ice crystals. At 10 km, the formation of new ice crystals is more
  unlikely than at 11.2 km, as most ice at this altitude likely originated from higher layers. Competition for available water
  vapor further reduces the potential for ice nucleation.





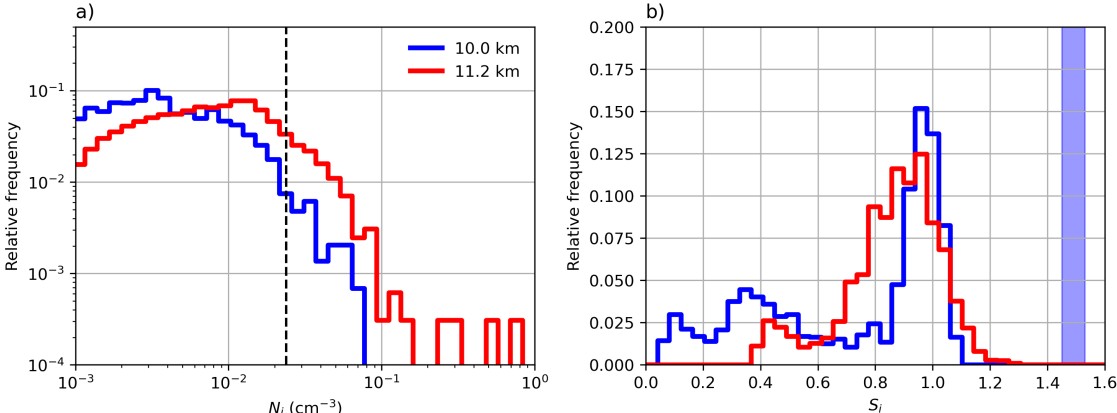

**Figure 8.** (a) Distribution of $N_i$ measured with 2DS inside cirrus clouds at two constant altitude levels with most continuous time series measured. Dashed line is drawn at the concentration of the mean clear-air concentration of mineral dust (0.0239 cm$^{-3}$) by PALMS. (b) Supersaturation over ice measured at two constant altitude levels. Blue shading indicates homogeneous freezing threshold $S_i$ between 210-230 K based on equation presented in (Ren and Mackenzie, 2005).

– Another critical factor is the entrainment of ice crystals by dry pockets of air. As ice crystals traverse, they can enter layers of air that are subsaturated with respect to ice ($S_i < 1$). In these subsaturated regions, some ice crystals sublimate, leading to a reduction in the $N_i$. This is more likely in the cirrus bottom altitudes (10 km) where the ice crystals have on average had a longer history.

## 4 Model setup

The UCLALES-SALSA simulations aim to attain results that closely align with measured data. The troposphere from the surface (0 km) to 14 km is simulated using a 3D domain to explore the impact of horizontal variability. The vertical resolution is set to $\Delta z = 300$ m below 6000 m and $\Delta z = 50$ m above 7000 m. Above 12500 m, the vertical resolution is lowered linearly to $\Delta z = 300$ m. The resolution is linearly decreased between 6000 and 7000 m. The vertical grid points are initialized by reading from file `zm_grid_in`. The time step is capped at a maximum of $\Delta t = 1$ s, which is suitable for simulating cloud microphysics. Data output is recorded at intervals of 5 minutes. The domain size spans $3 \times 3$ km with a horizontal resolution of $\Delta x = \Delta y = 50$ m in each direction. This resolution enables the simulation of small-scale eddies (Bryan et al., 2003). Additionally, sensitive tests are done by switching off ice nucleation processes and radiation scheme, and also running a 1D column simulations to examine horizontal variability free conditions. The meteorological variables used for the initial profile come from combined products of WB-57F measurements during the first descend towards the cirrus cloud layers before 18 UTC and the ERA5, shown in Fig. 9. The measurements made during this descend provides information from meteorological conditions throughout the cirrus clouds altitudes from 9 to 15 km. Data from ERA5 is used for the profile below 9 km as the





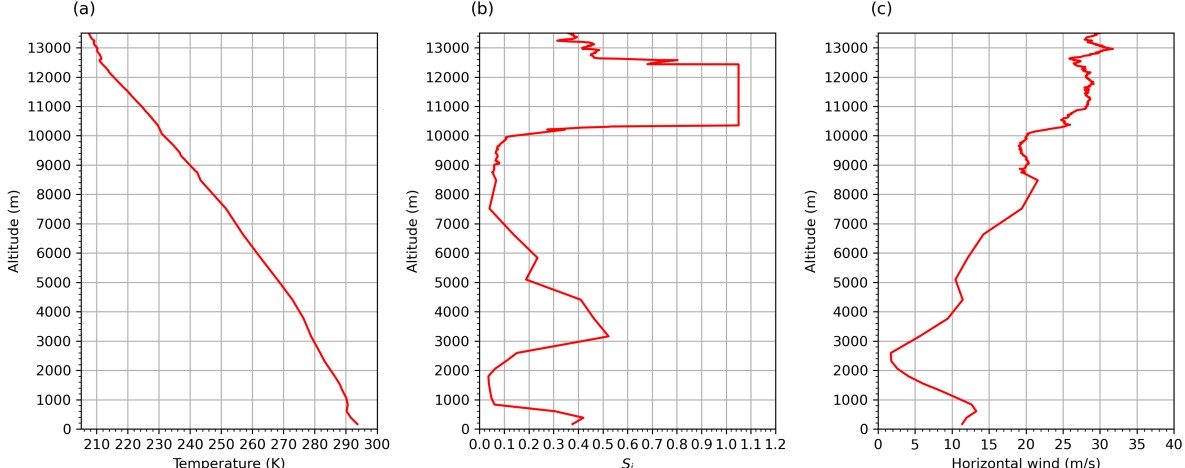

**Figure 9.** Initial profile temperature (a), $S_i$ (b) and horizontal wind used for the simulations (c).

aircraft did not descend below that altitude. It is, however, concluded that the lower resolution of the ERA5 is sufficient for
225  lower altitudes as the dynamical effects of lower atmosphere has limited effect on the cirrus cloud layer as the atmosphere is
relatively stable throughout. The stability within the cirrus cloud forming layer (hereby also known as supersaturated layer)
was relatively high and convection can not be expected. The temperature and humidity fields are perturbed with standard
deviations of approximately $\sigma_\theta = 0.025$ K and $\sigma_q = 2 \cdot 10^{-6}$ kg kg$^{-1}$. These values are in the same order of magnitude to the
values measured with WB-57F inside high humidity layers. The supersaturated layer with relatively high supersaturation over
230  ice is set between $10 < z < 12.1$ km. The initial $S_i$ in these levels was set to $S_i = 1.05$, approximately corresponding to the
equilibrium humidity levels typically observed in fully matured cirrus clouds. Satellite imagery revealed that new cirrus clouds
tended to form or intensify predominantly in regions where existing cirrus clouds were already present. This suggests that the
mean atmospheric conditions in areas with sufficiently high $S_i$ for cirrus formation were likely close to dynamic equilibrium.
To simulate the effects of large-scale forcing observed in back-trajectory analysis, a constant updraught is applied to the model
235  domain between altitudes of 3 km and 13 km. Constant updraught values of $w_{\mathrm{LS}} = 2, 3, 4, 5$ cm s$^{-1}$ is used to simulate the
effects of varying magnitude on the ice nucleation. After the model domain is lifted by 300 m, the ascent motion is halted. The
300 m lift corresponds to approximate maximum perturbation observed in the back-trajectory analysis. The lifting of the air
leads to cooling with temperature rate given by the following equation:

$$\frac{dT}{dt} = \frac{dT}{dz}\frac{dz}{dt} = -\frac{g}{c_p}w = -\Gamma \cdot w \tag{1}$$

240  where $\Gamma$ denotes the dry lapse rate, $g$ and $c_p$ denote the gravitational acceleration and the specific heat capacity for constant
pressure. The cooling in turn increases the $S_i$ as colder air has lower capacity to hold humidity. The initial choice of $S_i$
significantly influences how high $S_i$ can rise during a limited updraught duration. For instance, with a lapse rate of $\Gamma = 10$
K/km, the maximum cooling achieved with 300 m of lift corresponds to a temperature decrease of approximately 3 K. This,





in turn, leads to an estimated increase in supersaturation of $\Delta S_i \approx 0.45$. The $S_i$ levels after 300 m of lift correspond to level required for homogeneous freezing.

Additionally, the model domain experiences horizontal fluctuations in temperature, humidity and wind, and thus they are sensitive to more than the large scale forcing alone. For that, sensitivity runs are done without large scale forcing, radiation scheme and ice nucleation. Each simulation has 4 hour spin-up period where cloud microphysics and large scale forcing are disabled. The length of the spin-up was decided based on the relaxation of turbulent kinetic energy around 4 hours into the simulations.

Ahola et al. (2020) implemented various freezing mechanisms, however, in this study, mainly homogeneous freezing and deposition nucleation are turned on as the typical temperatures within upper tropospheric cirrus clouds are below -38° where pure water does not stay in liquid state. By default, UCLALES-SALSA uses homogeneous and heterogeneous ice nucleation parametrization schemes based on Khvorostyanov and Curry (2000), however, due to the heterogeneous ice nucleation schemes being stochastic (time dependent), it was concluded that ice nucleation could be greatly over-estimated in this particular study. To overcome this issue, a deterministic, time-independent deposition nucleation parametrization developed by Ullrich et al. (2017) for uncoated mineral dust particles was implemented to SALSA. The reparametrization was created by using a fit to ice nucleation activity of several mineral dust particles presented in Kanji et al. (2011). For this scheme, tracking of activated INP fractions is necessary since deterministic parametrizations base their ice nucleation activity on original INP population. Homogeneous freezing is implemented based on the temperature and $S_i$ relation presented in (Koop et al., 2000).

The aerosol profiles for insoluble and soluble aerosols are derived from FCAS II and PALMS measurements. The analysis of the data was done using the method described in Froyd et al. (2019). In this case study, mineral dust is treated as the only insoluble aerosol particle or heterogeneous INP, while other insoluble aerosols were possibly present in aerosol population such as soot from biomass burning (34%) were excluded from heterogeneous INP population. Large number of sulfates, organics and nitrates (26%) were present in the aerosol population, particles which most likely act as INPs in homogeneous freezing.

The measured mineral dust concentration on April 16 in the upper troposphere (with lower stratospheric air filtration) was $2.39 \times 10^{-2}$ cm$^{-3}$. These values are slightly lower than the MACPEX campaign average, however, in line with expected range of values in low latitude Northern hemisphere measured during ATom campaigns presented in Froyd et al. (2022). Additionally, this concentration is within the critical concentration for the heterogeneous ice nucleation to dominate over homogeneous freezing as reported in previous studies (Gierens, 2003; Spichtinger and Gierens, 2009). The UCLALES-SALSA is run with two different concentrations of mineral dust: at the measured concentration and at a reduced concentrations, corresponding to 0.1 of the measured concentration. The reduced concentration represents a case where the mineral dust has lower INP activity due to coating, or has been scavenged by previous cirrus formation events. These runs are referred to as STND (standard) and AGED respectively from hereafter.

On the other hand, the rest of aerosols measured are treated as soluble sulfate particles (H$_2$SO$_4$) and they are allocated to a-bins which potentially nucleate via homogeneous freezing from aqueous soluble droplets. The concentration of sulfates is set at constant $33.1 \times 10^{-2}$ cm$^{-3}$ based on the measured values for sulfates, organics, and other aerosol. The aerosols of both a- and b-bins are distributed into four modes of log normal distributions and the parameters are shown in Table 1. Other





possible insoluble aerosol such as black carbon are not included in the simulations. Once ice nucleation occurs in SALSA,

|  | $D_{g,1}$ | $\sigma_1$ | $N_{t,1}$ | $D_{g,2}$ | $\sigma_2$ | $N_{t,2}$ | $D_{g,3}$ | $\sigma_3$ | $N_{t,3}$ | $D_{g,4}$ | $\sigma_4$ | $N_{t,4}$ |
|---|---|---|---|---|---|---|---|---|---|---|---|---|
| Sulfates | 0.118 | 1.19 | 27.3 | 0.183 | 1.3 | 5.76 | 0.444 | 1.29 | 0.049 | 0.722 | 1.13 | 0.010 |
| Mineral dust | 0.207 | 1.3 | 0.0178 | 0.222 | 1.04 | 0.363 | 0.476 | 1.09 | 0.0033 | 0.737 | 1.13 | 0.0003 |

**Table 1.** Parameters of the four log normal distributions. Geometric mean diameter $D_g$, geometric standard deviation $\sigma$ and mode total number concentration $N_t$ in cm$^{-3}$.

ice crystals grow via a vapour deposition scheme according to formulation by Jacobson (2005). The ice crystal bins represent ice crystal mass sizes ranging from 2 to 500 $\mu$m. For processes such as sedimentation, condensation, and coagulation, only ice-related particle processes are modelled, with an exception on condensation of water vapour on aerosol particles to enable homogeneous freezing. Ice crystals are assumed to be spherical for sizes below 40 $\mu$m and bullet rosettes for larger particles. The habit dependence on size roughly follows the suggestion by Baum et al. (2005), which indicates that ice crystals below 60

$\mu$m are droxtals, while larger particles exhibit more complex habits. Images captured with the 2DS instrument during the April 16th flight show that a significant portion of larger ice crystals were bullet rosettes. To include the habit dependence on the crystal size, modification to mass transfer parametrizations were made. The capacitance was modified to reflect the increase in complexity of the ice crystal habit for larger crystals:

$$C_i(D_i) = \begin{cases} 0.5 D_i, D_i < 40 \, \mu\text{m} \\ 0.25 D_i, D_i \geq 40 \, \mu\text{m} \end{cases}$$

, where $D_i$ is the diameter maximum size of an ice crystal. Additionally, the ice crystal shape parameters (mass-diameter and area-diameter relationship parameters) were modified to accommodate the habit dependency by crystal size and are given in Table 2. SALSA calculates terminal velocity of ice crystals by the formulation described in Mitchell and Heymsfield (2005) (previous versions of SALSA use the Khvorostyanov and Curry (2002) formulation).

In addition, the existing vapor mass flux equation inside SALSA was modified to include the mass accommodation coefficient

|  | $\alpha$ | $\beta$ | $\gamma$ | $\sigma$ |  |
|---|---|---|---|---|---|
| Spherical | $\frac{\pi}{6}\rho_P$ | 3 | $\frac{\pi}{4}$ | 2 | $D_i < 40 \, \mu$m |
| Bullet rosette (5-sided) | 0.0138 | 2.26 | 0.2148 | 1.7956 | $D_i \geq 40 \, \mu$m |

**Table 2.** Values for parameters in mass-diameter and area-diameter relationships. The relationship between mass and diameter is given as $m = \alpha D_i^{\beta}$. The projected area and diameter relationship is given as $A = \gamma D_i^{\sigma}$.

for water vapor uptake of ice crystals ($\alpha_c$) and it is set at $\alpha_c = 0.5$ for this case study. This value has a large uncertainty currently with measurements showing values in range $0.004 \geq \alpha_c \geq 1.06$ (see, Skrotzki et al., 2013). By default the $\alpha_c$ was practically set at 1 previously (with no parameter described), however, this value falls in the high end of the typically observed values.





## 5 Comparison of simulation data to observations

### 5.1 The effect of aged mineral dust on ice nucleation activity

In Fig. 10, the time evolution of a cirrus cloud deck produced with constant updraughts of $w_{\mathrm{LS}} = 2 - 5$ cm s$^{-1}$ is shown for two simulation set ups STND (runs with PALMS concentration of mineral dust) and AGED (runs influenced by ageing of mineral dust), of which we focus first on the STND cases (left panels). The cirrus cloud initially forms in upper parts of the supersaturated layer, above 11 km, due to the high efficiency of heterogeneous ice nucleation at colder temperatures. Ice

nucleation also becomes efficient in the lower parts of the supersaturated layer (below 11km) as temperatures continue to drop and $S_i$ increases. Once the constant updraught ceases, new ice crystal formation largely subsides. This happens due to the nature of the deterministic freezing parametrization used in this study, which does not allow further ice nucleation in constant $S_i$ and warming temperatures. The combination of heterogeneous ice nucleation and vapour deposition growth of ice crystals effectively suppresses the increase in $S_i$, preventing it from reaching the levels required for homogeneous freezing. Even

with an updraught of $w_{\mathrm{LS}} = 5$ cm s$^{-1}$ in Fig. 10)g, where growing ice crystals have less time to consume water vapour, the suppression is strong enough to inhibit homogeneous freezing. This indicates that the measured mineral dust concentration is sufficiently high to prevent competition between homogeneous and heterogeneous ice nucleation mechanisms. If the constant updraught had continued for a longer period, homogeneous freezing could have occurred due to the depletion of available mineral dust particles in the supersaturated layer. However, back-trajectories suggested that the maximum continuous large

scale shifts of supersaturated layer was not sufficient for this to occur.

The large-scale forcing also influences $N_i$ indirectly. When $w_{\mathrm{LS}}$ increases, the activation of mineral dust is more efficient and the $S_i$ suppression is weaker due to ice crystals having less time to consume water vapour, resulting in higher $N_i$ within a relatively short time frame. Conversely, in cases of slower updraughts, the activation process is more gradual, leading to lower $N_i$ overall.

Looking at AGED cases (right panels in Fig. 10), initially, only few ice crystals are produced as the number of ice-nucleation-active mineral dust particles is low. The suppression of $S_i$ by the few heterogeneous ice crystals growing is not sufficient and homogeneous freezing occurs. In Fig. 10 right panels, the occurrence of homogeneous freezing is noticeable from high gradient of $N_i$ as the homogeneous freezing occurs only when critical $S_i$ for homogeneous freezing is reached and produces large number of ice all at once. Interestingly, homogeneous freezing occurred at two distinct altitude ranges: between 10.5–11.8

km and within a narrow layer around 12 km. This is evident from the sharp gradient in $N_i$ shown in Fig. 10. Near the top of the cirrus cloud (around 12.5 km), homogeneous freezing is facilitated by the fact that heterogeneous ice crystals deplete less water vapour, leaving more available humidity compared to the lower layers, where larger ice crystals have grown and consumed more moisture. In the 10.5–11.8 km range, sufficient amplitude fluctuations in $S_i$ can occasionally push supersaturation beyond the threshold required for homogeneous freezing, even in the presence of heterogeneous ice. This phenomenon is discussed further

in Section 5.4.



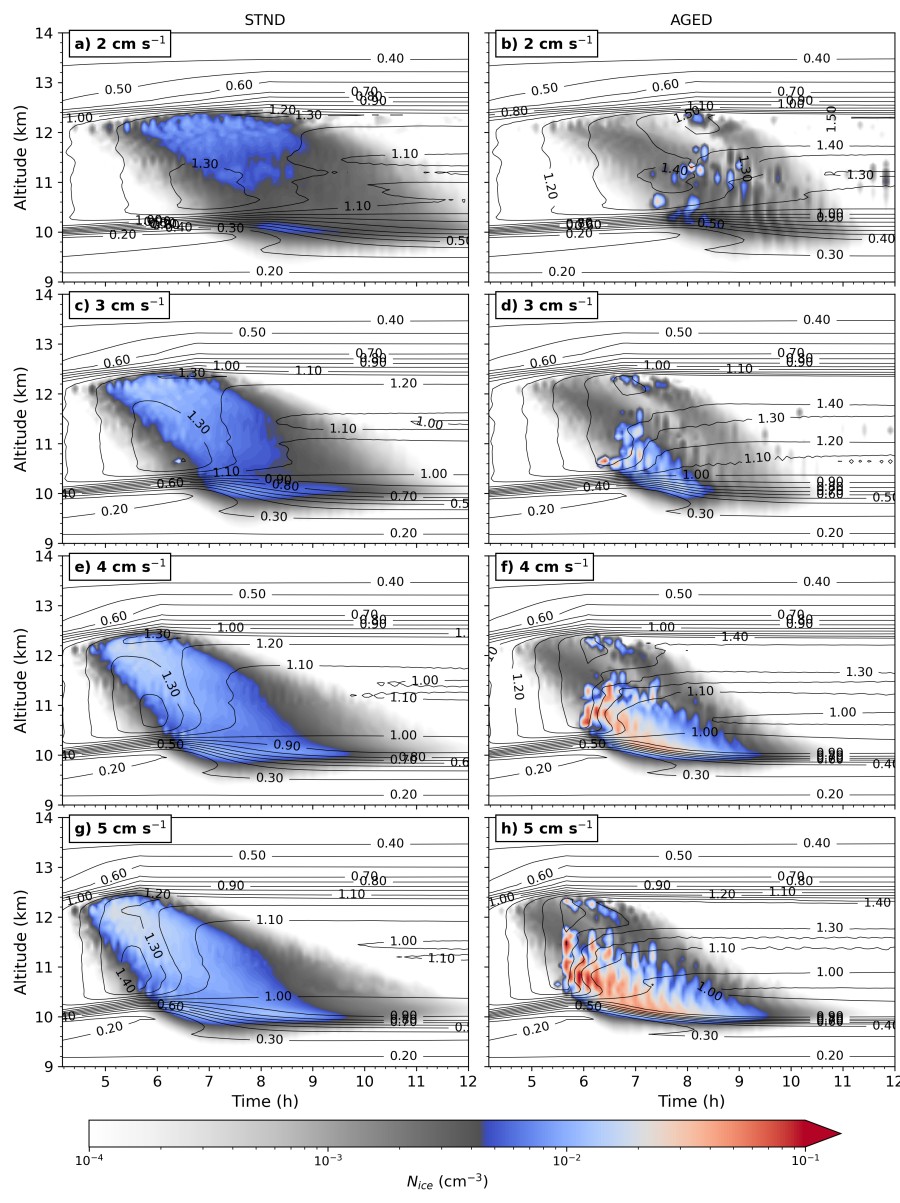

**Figure 10.** Cirrus clouds simulated with $w_{LS} = 2 - 5 \, \text{cm} \, \text{s}^{-1}$ for measured mineral dust concentrations (STND; a,c,e,g) and reduced mineral dust concentrations (AGED; b,d,f,h). Black contour lines represent $S_i$ at 0.1 intervals. The colour map highlights $N_i$ at critical concentration levels relevant to this study: blue shades correspond to values typically associated with heterogeneous ice nucleation, while red shades indicate concentrations consistent with homogeneous freezing. Grey shades separate blue values from red, as the analysis focuses on high $N_i$ levels in blue and red ranges. The profiles show a 1D vertical slice at the edge of the model domain, with $S_i$ contours representing horizontal averages to reduce noise from horizontal fluctuations.



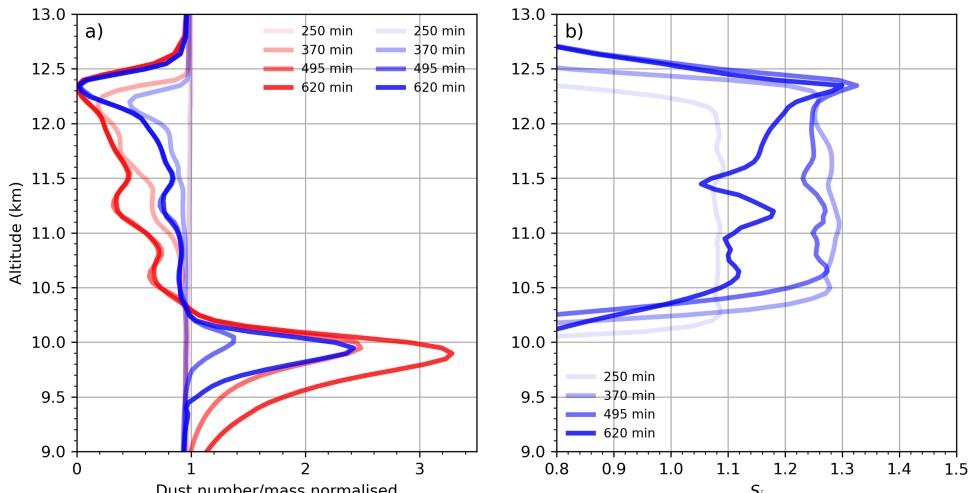

**Figure 11.** Time series of the mineral dust number/mass (blue/red solid lines respectively) concentration (a) and $S_i$ profiles (b) from the beginning to the end of the simulation using the STND setup. The mineral dust data is presented as normalized dust number/mass, expressed as a fraction of the initial values.

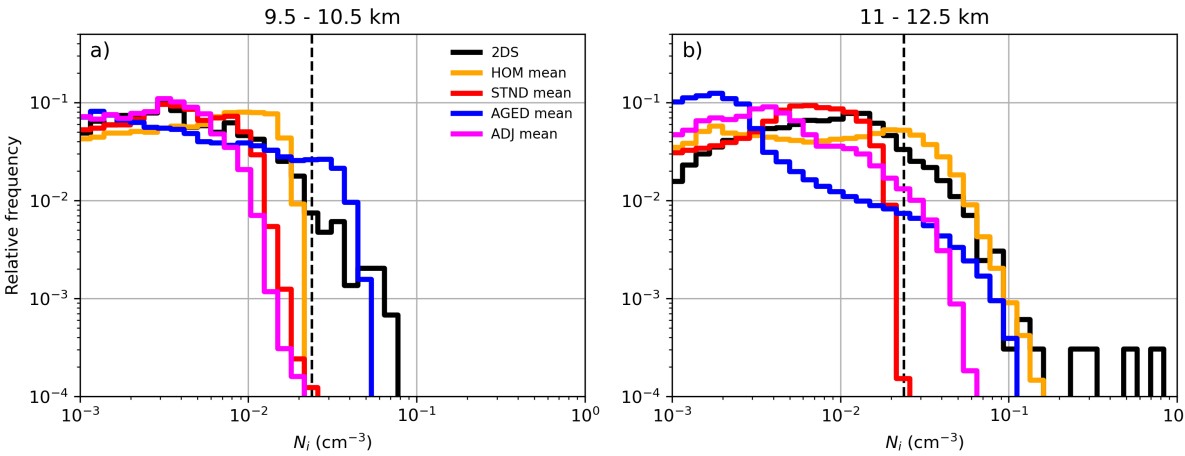

**Figure 12.** Distribution of $N_i$ in two altitudes that the WB-57F measured continuous statistical data. The measurement data is from 2DS instrument which is in solid black lines. The STND and AGED runs stand for standard with measured mineral dust concentration, AGED for mineral dust particles that have aged. ADJ stands for adjusted profile set up runs and HOM for homogeneous freezing only runs. PALMS measured mineral dust concentration shown with black dashed vertical line at $2.39 \times 10^{-2}$ cm$^{-3}$.

A notable pattern emerges from the model simulations, as shown in Fig. 11a, where mineral dust particles are depleted in both number and mass due to heterogeneous ice nucleation. The most efficient removal of dust occurs between 12 and 12.5 km,



where only a small fraction of mineral dust particles remains by the end of the simulation. Below 12 km, a significant difference is observed between the depletion of dust number concentration and mass concentration, with mass being more heavily depleted. This discrepancy is largely attributed to the preferential activation of larger INPs, which possess greater surface areas and more active ice-nucleating sites, as described by the Ullrich et al. (2017) parameterization.

In contrast, the total number of dust particles remains relatively unaffected compared to the total mass, as the activated particles are predominantly larger dust particles. While these larger particles are less numerous, they play a critical role in ice nucleation under the temperature ranges considered in this study. Between 12 and 12.5 km, the activation of mineral dust particles appears to be less dependent on particle size, as temperatures are below 215 K. For the Ullrich et al. (2017) parameterization, the efficiency difference between particle sizes becomes more pronounced in the temperature range of 215–230 K.

The Figure 11a additionally shows that the activated mineral dust particles eventually sediment to altitudes below saturation over ice (Fig. 11b). As a consequence of cirrus formation, mineral dust particles and other INPs involved can be transported vertically significant distances.

These findings have important implications for interpreting the mineral dust concentration measured by PALMS. The clear-air concentration and size distribution of mineral dust may reflect a state influenced by previous nucleation events that occurred before the measurements. Most of the dust measurements were taken at constant altitudes of 10 and 11.2 km, are not necessarily representative of ice initiation, which for these clouds is at >12 km. While there may be a lack of larger dust particles at these altitudes, their contribution to the overall number concentration would be minimal, and thus not easily detectable.

Additionally, ice crystal vapour growth plays a crucial role in controlling the vertical distribution of humidity as shown in right panel of Fig. 11b. As ice crystals grow and sediment through the supersaturated layer, they deplete the surrounding humidity. Consequently, the vertical distribution of $S_i$ exhibit a distinct pattern: near the top of cirrus clouds, $S_i$ remains significantly above saturation, while below, it approaches saturation. This occurs because the descending ice crystals progressively consume water vapour during sedimentation.

Figure 12 presents the distribution of $N_i$ at two different altitude levels based on 2DS measurements and model outputs. The maximum $N_i$ did not surpass the mineral dust concentration with the STND runs, emphasizing that with a substantial presence of mineral dust throughout the supersaturated layer, homogeneous freezing would be suppressed. The similar findings were reported in Spichtinger and Gierens (2009) where they simulated ice nucleation using different concentration levels of heterogeneous INP and found that heterogeneous ice nucleation dominates above $20\,\mathrm{L}^{-1}$ which is in this case study right below the measured concentration of dust particles.

In contrast, the AGED runs demonstrated that the $N_i$ surpassed the PALMS measured mineral dust concentration. The $N_i$ values in the lower parts of the cirrus clouds show a relatively good agreement with the 2DS observations. However, in the upper levels, a significant portion of $N_i$ is focused around $10^{-3}\,\mathrm{cm}^{-3}$. Higher $N_i$ values were predominantly generated by homogeneous freezing. At lower levels (9.5-10.5 km), homogeneous freezing seemed to generate higher $N_i$ much more frequently. This increased homogeneous freezing in the lower parts of the cirrus clouds can be explained by the presence of high-frequency gravity waves in these layers, which provided the necessary conditions for more frequent nucleation events.



## 5.2 Accounting for prior cirrus formation

The previous runs suggested that prior nucleation events likely removed a significant fraction of ice-nucleation-active mineral dust or other INPs from the supersaturated layer (Fig. 11a). To explore this further, additional simulations (referred to as ADJ runs) were conducted using adjusted dust and humidity profiles to approximate post-nucleation conditions, as shown in Fig. 11 for the dust profile. The $S_i$ profile was modified to reflect depletion of humidity following prior freezing events, resulting in a peak $S_i$ near the top of the supersaturated layer and reduced $S_i$ below it.

Direct validation of the elevated $S_i$ values by WB-57F measurements was not possible due to limited sampling, as the aircraft only recorded two passages through the upper cirrus layers, leaving no continuous dataset. Likewise, validation of the adjusted dust profiles was challenging, as the PALMS measurements were conducted at levels where the activation of mineral dust in absolute numbers proved to be not as efficient as in even higher altitudes. The measurements near the top of the cirrus were absent, similarly to the humidity data.

Nevertheless, GOES-16 imagery (Fig. 6 and ERA5 back-trajectories (Fig 5) indicate that the air mass had undergone intense ice nucleation events before reaching the measurement locations. This supports the hypothesis that active dust particles were transported downward, below the supersaturated layer, along with sedimenting ice crystals. Furthermore, there was no evidence of significant horizontal mixing within the cirrus layer that could have reintroduced ice-nucleation-active dust particles into the supersaturated region.

Figure 13 shows the cirrus clouds generated with the adjusted set up. Just as in the AGED runs, the cirrus undergoes two distinct nucleation events where significantly smaller number of ice is produced initially in the upper parts of cirrus (coloured in grey colors in Fig. 13 left panels), followed by nucleation events where a large quantity of ice is produced at once by homogeneous freezing.

Runs with heterogeneous ice nucleation turned off (HOM; Fig. 13 right panels) and were conducted as a reference case where

the remaining ice-nucleation-active mineral dust particles were largely absent from the supersaturated layer, likely depleted during previous ice nucleation events where $S_i$ had reached levels with high heterogeneous ice nucleation efficiency. The additional assumption is that the remaining mineral dust population is fully activated, corresponding to a frozen fraction of 100%. The homogeneous freezing occurred with 2-5 cm s$^{-1}$ updraught velocities around the time when $S_i$ reaches critical threshold $S_i$ (Fig.13 right panels). As there is no heterogeneous ice nucleation to suppress the $S_i$ prior to homogeneous

freezing, values of $N_i$ are at the highest possible out of all scenarios of model runs. Homogeneous freezing with high $N_i$ consumes efficiently humidity and the conditions inside cirrus clouds reach much faster the equilibrium state than in the ADJ. Additionally, the lifetime of cirrus are extended slightly as the homogeneously frozen ice crystals grow to smaller sizes and sediment slower as the consumption of water vapour immediately after ice nucleation is significantly larger in quantity.

## 5.3 Simulated ice crystal concentration suggests homogeneous freezing limited by initial heterogeneous ice nucleation

In the ADJ cases, the $N_i$ exceed the dust concentration at least in the upper parts of the cirrus at altitudes 11-12.5 km (12). This is due to the low number of dust in the upper parts of the cirrus, enabling homogeneous freezing.





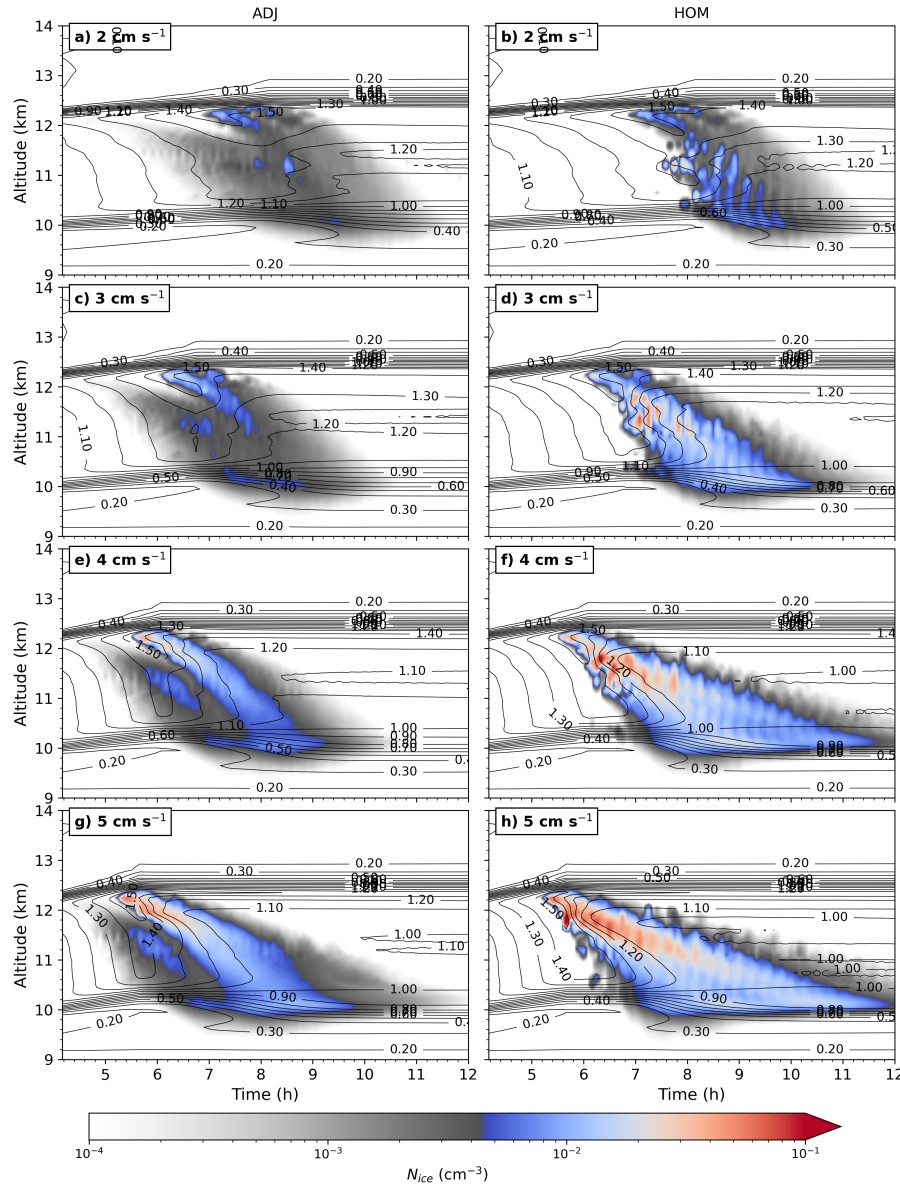

**Figure 13.** Cirrus cloud produced with $w_{LS} = 2 - 5$ cm s$^{-1}$ with adjusted mineral dust concentration and humidity (a,c,e,g) and only homogeneous freezing switched on (b,d,f,h). The explanation of the details on the figure, see Fig.10.

There was a significant absence of high $N_i$ in the 9.5-10.5 km altitudes for ADJ campared to AGED. This is mainly due to the $S_i$ being lower from the start and the lower parts of cirrus clouds and higher number of mineral dust particles present, limiting the occurrence of homogeneous freezing. The high $N_i$ in AGED cases at altitudes 9.5-10.5 km was a result from uniformly

high $S_i$ throughout the supersaturated layer, combined with low number of ice nucleation-active mineral dust particles. In





Fig. 12, the homogeneously frozen $N_i$ distribution is shown. It seems that the shape agrees relatively well with the 2DS measurements, especially well at the high end of $N_i$. There is a slight lack of low $N_i$ in the upper parts of cirrus and the high frequency is concentrated in the narrow range of values above the mean of measurements. Among all cases in this study, the HOM simulations show the closest statistical resemblance to the 2DS measurements, proving that without any influence of

heterogeneous ice nucleation the $N_i$ exceed the clear air concentration of mineral dust most efficiently.

## 5.4  The effects of atmospheric fluctuations

The 3D simulations offered a comprehensive depiction of atmospheric variability in meteorological conditions. Despite relatively stable atmosphere within the supersaturated layer without presence of convection, small-scale waves resembling gravity waves emerged within the model domain. The vertical wind ($w$) in MMS measurements on April 16th and in model domain

are shown together in Fig. 14. The $w$ observed in the model domain stemmed from local instabilities arising from vertical wind shear at altitudes between 10 and 11 km. These waves induced notable fluctuations in $S_i$ (as shown in Fig.15), with variability around 0-5% throughout supersaturated layer during the spin-up phase of the simulations. Sensitivity tests were

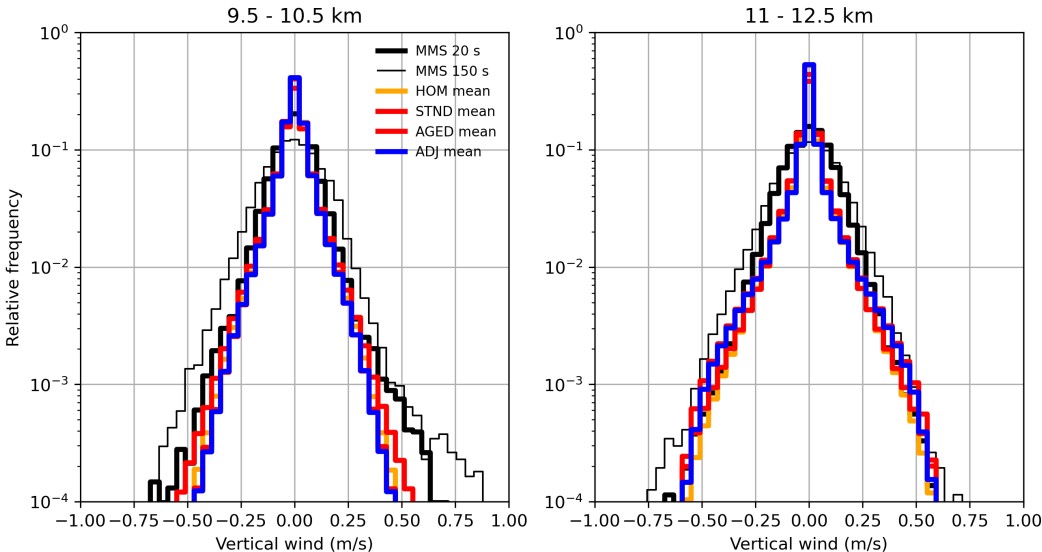

**Figure 14.** Frequency distribution of vertical wind in two altitudes for the MMS measured continuous statistical data with 20 and 150 second running mean filtering and model data for all of the simulation set ups.

done by adjusting the horizontal resolution, domain size in both X and Y directions. These waves were present even in larger domains excluding the possibility that these waves were predominantly standing waves that self-amplified over time due to

the lack of kinetic energy dispersion inside the domain (Fig. S1). By changing the horizontal grid resolution, the size of these waves slightly changed, and increasing the grid size above $\Delta x = 100$ m diminished the presence of these small scale waves significantly.





Examining Fig. 14, it becomes evident that the measured MMS $w$ filtered with 20-second filter, the distribution is very closed to simulated $w$ distribution. The model and simulated $w$ both exhibit an ogival-shaped distribution similar to one reported by
Gierens et al. (2007). The similarity of the $w$ distributions confirm that the modelled fluctuations are simulated correctly. The modelled ice nucleation mechanisms do not seem to affect significantly the shape of the distribution, however, some variation is visible between each model run. Overall, the relatively good agreement of distributions mean that the variability in the model domain emulates the wave activity inside real atmosphere. The 150-second filtered MMS $w$ data (Fig. 14) broadened the distribution shape slightly as it includes more mesoscale waves of larger wavelengths that were not simulated inside $3 \times 3$
km model domain. The higher range of MMS $w$ aligns well with the modelled $w$, which is crucial since homogeneous freezing efficiency is highly sensitive to these magnitudes of $w$.

It was stated previously that the maximum $N_i$ achieved in HOM cases was not clearly correlated to the imposed large-scale $w$. The cooling within the supersaturated layer was primarily influenced by the large-scale $w$; however, small-scale turbulence induced local variations in temperature and humidity. These fluctuations caused the $S_i$ to intermittently reach the critical level
necessary for homogeneous freezing. While the large-scale $w_{\mathrm{LS}}$ ranged from 2 to 5 cm s$^{-1}$, turbulent $w$ fluctuated between $\pm 0.6$ m s$^{-1}$ as seen in Fig. 14, frequently exceeding the magnitude of the large-scale $w_{\mathrm{LS}}$. The high frequency of turbulence resulted in high $S_i$ with short lifetimes as shown in example in Fig. 15. Sensitivity tests conducted using a single-column setup (results in Supplementary data S8) reveal that, in the absence of horizontal variability and turbulence, high $N_i$ at the high tail end of distributions (in Fig. 12) are completely absent.

### 5.4.1   Practical examples of heterogeneous and homogeneous freezing cases

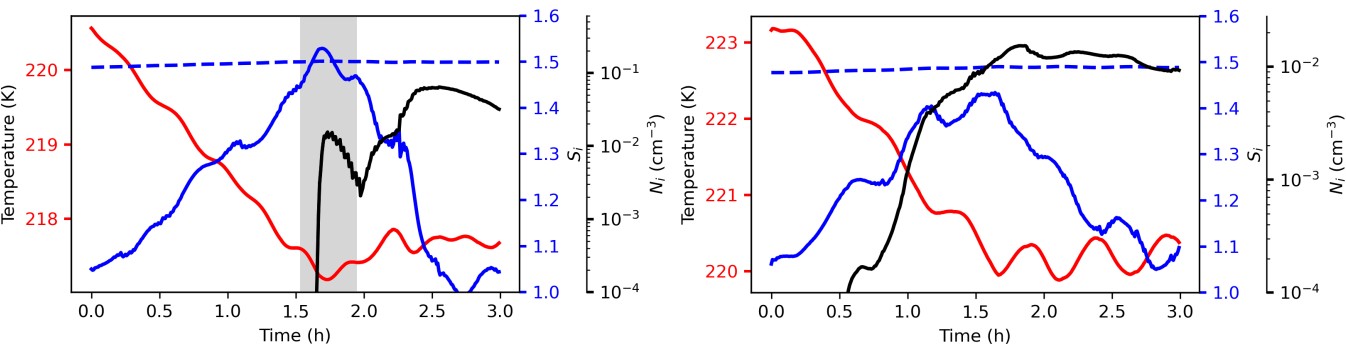

**Figure 15.** (a) Homogeneous freezing case from HOM run and (b) heterogeneous ice nucleation case from STND run. The plots show temperature (red), supersaturation over ice ($S_i$, blue), and ice crystal number concentration ($N_i$, black) before and during freezing events. The critical supersaturation for homogeneous freezing, based on Eq. 10 in Ren and Mackenzie (2005), is indicated by dashed blue lines. The timing of the homogeneous freezing event is highlighted with gray shading.





Figure 15a illustrates the meteorological variables immediately before and after the onset of homogeneous freezing with parcel trajectories. The trajectory reveals a relationship between turbulent fluctuations, $S_i$, and the $N_i$. It is evident that $S_i$ and temperature fluctuates apart from the general trend before the onset of homogeneous freezing, with these variations occurring

over phase times significantly smaller than the imposed vertical velocity which represents the behaviour of larger scale gravity wave. As $S_i$ increases and approaches the critical threshold $S_i$ for homogeneous freezing, $N_i$ rapidly reaches to peak levels. Immediately after a freezing event, the concentration starts to decrease quickly as the ice crystal fall out of the air parcel. Previous numerical studies (Jensen et al., 2010; Spichtinger and Krämer, 2013) have shown that the interaction between high-frequency gravity waves and slow, large-scale cooling can result in different ice concentrations compared to scenarios involving

only large-scale cooling. These interactions can lead to either higher or lower concentrations of ice crystals, depending on the phase of the gravity waves and the cooling rates at the time of homogeneous freezing. The homogeneous freezing event depicted in Fig. 15 represents a situation where the wave was in a cooling phase, resulting in a high $N_i$.

Figure 15b illustrates a case of heterogeneous ice nucleation. Each peak in $S_i$ along the parcel's trajectory corresponds to an increase in $N_i$, as new ice crystals form during these supersaturation events. However, after 1.5 hours into the simulation, this

correlation becomes less evident due to the influence of sedimenting ice crystals, which alter the local ice crystal population. As the cooling phase ends, $S_i$ begins to decline as the growing ice crystals deplete available water vapour. Concurrently, $N_i$ decreases gradually as ice crystals sediment out of the layer, while additional crystals are introduced into the population from sedimentation originating in higher altitudes. The presence of these growing ice crystals effectively prevents $S_i$ from reaching levels necessary for homogeneous freezing. This suppression becomes particularly pronounced around 1.5 hours into

the simulation, as the ice crystals efficiently limit further increases in $S_i$.

## 6 Possible limitations of this study

In this study, only mineral dust particles were considered as INPs. However, other aerosol components, such as soot, glassy particles, and coated soluble droplets, may also contribute to the heterogeneous INP population and influence the overall ice nucleation activity in cirrus clouds. The ice nucleation efficiency of these additional INP types remains less well understood,

and their inclusion in the model would introduce further uncertainties. Furthermore, the conclusions drawn from this study are constrained by the limited sample size of the measured aerosol particles and the restricted vertical sampling, which was conducted only between 10 and 11.2 km. As a result, this study cannot fully validate the simulation outcomes or confirm the robustness of its conclusions.

Furthermore, this study simulated cirrus cloud formation under specific conditions that allowed homogeneous freezing to oc-

cur in scenarios where the effects of heterogeneous ice nucleation were limited. If the model had been run under conditions unfavourable to homogeneous freezing, the results would have looked significantly different, with $N_i$ primarily constrained by the number of heterogeneous INPs. While simulating cirrus with lower $N_i$ might have increased the variability of cloud conditions, cirrus clouds with higher $N_i$ would still have dominated the analysis, meaning the overall conclusions would not have been significantly affected.




In contrast, adjusting the initial humidity or increasing the vertical displacement of the supersaturated layer would have favoured more efficient homogeneous freezing. However, the main interpretation of this study remains unchanged. The behaviour of freezing in cirrus clouds—and the occurrence of homogeneous freezing—was shown to be highly dependent on the presence of heterogeneous INPs. Homogeneous freezing did not occur, regardless of the initial $S_i$ or the magnitude of the vertical displacement, when the number of heterogeneous INPs at the cirrus cloud top matched the concentrations from the
PALMS observed at lower levels.

## 7 Conclusions

Based on the observations from the MACPEX campaign, the cirrus clouds on April 16th, 2011, were predominantly formed through homogeneous freezing, as evidenced by ice residual particle (IRP) analysis. Other days of the MACPEX campaign with similar conditions showed that the heterogeneous ice was dominating the IRP analysis. We investigated the role of het-
erogeneous ice nucleation with the UCLALES-SALSA model, and the results showed that previous events with heterogeneous ice nucleation increases the likelihood of homogeneous freezing occurring during subsequent ice nucleation events. Simulations with measured mineral dust concentrations (STND) showed an almost complete absence of homogeneous freezing. This suggests that prior heterogeneous nucleation events likely depleted the heterogeneous INPs from certain layers of the cirrus clouds, particularly in the colder upper regions. This depletion indirectly enabled the conditions necessary for subsequent ho-
mogeneous freezing to occur.

Finally, vertical shear instabilities within the supersaturated layer led to turbulence and gravity waves, which in turn caused substantial variability in temperature and $S_i$. This wave activity resulted in greater variability in the $N_i$ and the rate of homogeneous freezing compared to a scenario without such waves. Notably, the changes in temperature and $S_i$ were slower for large-scale motions, highlighting the significant role of smaller-scale perturbations in affecting ice nucleation processes.
Although clear-air INP measurements were not available at cloud top altitudes, MACPEX dust concentrations typically had little vertical structure throughout the cirrus regime. For the Apr 16 case we assume that dust measured at mid- and low-cloud levels was representative of initial cloud top conditions. This underscores the need for improved measurements of aerosol populations, with high sample rate of INPs in both horizontal and vertical directions in future campaign. This study also demonstrated that high resolution three dimensional LES model studies are able to simulate huge variability of $N_i$ inside cirrus
that large scale relatively low resolution models simulating the global impacts of cirrus clouds struggle. The LES approach demonstrated that small scale gravity waves can be simulated without using a separate parameterization to simulate the effects of small scale gravity waves (Jensen et al., 2013a). In future, inclusion of more INPs to simulate cirrus clouds could clear uncertainties that were not explored in this study.

*Code availability.* The source code of the model UCLALES-SALSA is available from GitHub at https://github.com/UCLALES-SALSA/
UCLALES-SALSA under release tag MACPEX_icenucl.



*Data availability.* The campaign data from the MACPEX field study, which was used in this study, is publicly available at the following URL: https://espoarchive.nasa.gov/archive/browse/macpex/WB57. UCLALES-SALSA model data for runs with single seed used for analysis archived at Zenodo (DOI:10.5281/zenodo.14500482. For whole dataset, contact the corresponding author.

*Author contributions.* KJ made the simulations, analysed the simulation results and wrote major parts of the text. CW contributed to ex-
periment design, size distribution analysis, and provided input on the manuscript. KF provided detailed analysis of PALMS spectra and guided with MACPEX campaign data. JD provided information on interpreting MMS instrument data and analysis of vertical wind data. AL supervised the project and provided input on the manuscript.

*Competing interests.* Some authors are members of the editorial board of Atmospheric Chemistry and Physics.

*Acknowledgements.* This work was supported by the Academy of Finland Flagship ACCC (grant no. 337552) and MEDICEN project (grant
no. 345125 and 359892). Supercomputing resources were provided by CSC– IT Center for Science, Ltd., Finland. We thank J.C. Wilson for use of the FCAS aerosol data.



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
