# Peer review of "Prior heterogeneous ice nucleation events shape homogeneous freezing during the evolution of synoptic cirrus"

_EGUsphere, 2025_

## Referee Comment (RC1)

**Prior heterogeneous ice nucleation events increase likelihood of homogeneous freezing during the evolution of synoptic cirrus**

Kasper Juurikkala et al.

**General:**

This manuscript examines the formation mechanism of a synoptic cirrus observed during the MACPEX field campaign. The observations suggest homogeneous freezing as the dominant nucleation mechanism, which is investigated through simulations with the UCLALES-SALSA model.

Generally, the manuscript deals with the topic of the influence of heterogeneously freezing particles (so-called INPs, mineral dust in this case) on the homogeneous freezing process. The theory of the microphysics of this process is well understood and the study does not provide new microphysical insights. What is presented are sensitivity studies with a model in which the measured properties of a cirrus, namely that it is formed by pure homogeneous freezing, can be confirmed by theoretical considerations. That means in summary the agreement between theory and observations is shown. Specifically, the study shows that pure homogeneous freezing took place in the upper layers of the observed cirrus, because the previously existing INPs had already frozen out earlier.

The study also examines in general the influence of INPs on the formation process - heterogeneous or homogeneous - of cirrus clouds in the specific dynamic situation of the observed cirrus clouds. The influence of small-scale fluctuations of vertical velocity is also discussed.

Overall, the study is interesting and timely and the methods used are appropriate, making the manuscript suitable for publication in ACP.

**(G 1)** However, I find the title a bit exaggerated and not quite appropriate for the study, even though I understand that the title is intended to arouse curiosity. I suggest toning down the title to something like

'Prior heterogeneous ice nucleation events shape homogeneous freezing during the evolution of synoptic cirrus'
or
'Prior heterogeneous ice nucleation events enable / allow pure homogeneous freezing during the evolution of synoptic cirrus'

or, completely different: 'The influence of heterogeneous ice nucleation on cirrus evolution: a case study of an observed homogeneously formed cirrus'

which to my feeling better reflect the content of the study. This content could/should also be better elaborated in the paper.

**(G 2)** In general, I have to say that the study could/should be significantly improved in terms of writing. In its current form, the reader cannot follow the content fluently - it took me a long time to put the puzzle together, and if I hadn't had to review the paper, I certainly wouldn't have read it to the end....

Often  (a) references to previous work is missing;  also,

(b) the model description is too brief.

For more detail see the see specific comments.

Further, (c) the manuscript is not well structured, which makes it unnecessarily difficult to understand. As outlined  in the specific comments, I recommend to

  – distribute Section 5.4  to Section 4 and  Sections 5.2/5.3  and
  – move Fig. 12  behind Figs. 11 and 13.

This would give the manuscript a clearer structure and Fig. 12 - the summary of the results - would  be  the final figure  to the paper, which I think is a better place then now in between more detailed results.

A further  idea might be to then    – merge sections 4 and  2.3  to  have only one section describing the model and the simulated scenarios.

My final assessment is that the paper needs fundamental revision,  in terms of the focus and findings of the study, the introduction to the topic, the incorporating of previous work and a more fluent structure.

**Specific comments:**

**(S 1) Line 33f:**  '*Over the past few decades, several key measurement campaigns (e.g., Krämer et al., 2009; Voigt et al., 2017) have been conducted in the UTLS.*'

Please add more recent work here:
- Krämer et al., 2009  reported multiple campaigns, the following studies  could be added:
 Krämer et al., 2016 (ACP),  Krämer et al. 2020 (ACP), Patnaude et al., 2021 (ACP), Ngo et al., 2024 (ACP).

Voigt et al. (2017) presents a single field campaign (ML-Cirrus), the following studies  could be  added:
i.e. Pan et al. (2010) (START08, BAMS),  Wendisch et al. (2016) (ACRIDICON-CHUVA, BAMS), Jensen et al. (2017) (ATTREX, BAMS),  Pan et al. (2017) (CONTRAST, BAMS).

These campaigns are included either  in  Krämer et al. (2020) or in  Ngo et al. (2024) (or both).

**(S 2) Line 38: '***Synoptic cirrus clouds primarily form through two dominant mechanisms: heterogeneous and homogeneous freezing.***'**

This applies not only to synoptic, but to all cirrus.

**(S 3) Line 38ff:** *'Heterogeneous ice nucleation …     In contrast, homogeneous freezing ...'*

Please provide references for heterogeneous and homogeneous freezing**.**

**(S 4) Line 48ff:** **'***Ice nucleation in cirrus clouds is strongly influenced by the abundance of INPs, which regulate how efficiently heterogeneous ice nucleation can suppress homogeneous freezing and activity.'*

Please provide references for the competition between heterogeneous and homogeneous freezing, i.e. influence of INP number and vertical velocity on the onset of homogeneous freezing. however, the present study lacks references to previous work. Here are some examples, but the list is not exhaustive.

– Spichtinger, P., and D. J. Cziczo (2010), Impact of heterogeneous ice nuclei on homogeneous freezing events in cirrus clouds, J. Geophys. Res., 115, D14208, doi:10.1029/2009JD012168.

– Rolf, C., Krämer, M., Schiller, C., Hildebrandt, M., and Riese, M.: Lidar observation and model simulation of a volcanic-ash-induced cirrus cloud during the Eyjafjallajökull eruption, Atmos. Chem. Phys., 12, 10281–10294, https://doi.org/10.5194/acp-12-10281-2012, 2012.

**–** Krämer, M., Rolf, C., Luebke, A., Afchine, A., Spelten, N., Costa, A., Meyer, J., Zöger, M., Smith, J., Herman, R. L., Buchholz, B., Ebert, V., Baumgardner, D., Borrmann, S., Klingebiel, M., and Avallone, L.: A microphysics guide to cirrus clouds – Part 1: Cirrus types, Atmos. Chem. Phys., 16, 3463–3483, https://doi.org/10.5194/acp-16-3463-2016, 2016.

– Kärcher, B., Jensen, E. J., & Lohmann, U. (2019). The impact of mesoscale gravity waves on homogeneous ice nucleation in cirrus clouds.
Geophysical Research Letters, 46, 5556–5565. https://doi.org/10.1029/2019GL082437

The authors miss to discuss this paper, although it also compares MACPEX observations with simulations.

– Kärcher, B., DeMott, P. J., Jensen, E. J., & Harrington, J. Y. (2022).
Studies on the competition between homogeneous and heterogeneous ice nucleation in cirrus formation. Journal of Geophysical Research: Atmospheres, 127, e2021JD035805. https://doi.org/10.1029/2021JD035805

**(S 5) Section 2.3 UCLALES-SALSA**

**a)** The information about the model is very limited. I recommend a short description of which parameterizations for heterogeneous and homogeneous freezing are used and how the processes (ice particle growth, evaporation, sedimentation, aggregation, ...) are treated in the model.

**b)** Further, please provide information about the sizes of the ice bins.

**(S 6) Figure 4**: What time resolution do the measurements shown have?

Also, why not showing the IWC (ice water content) in addition ? There was a total water instrument on board (CLH), so together with the HVW gas phase water you have a good IWC information.

**(S 7) Figures 2, 5, 6:**

To give the reader a better overview, I recommend making one plot from Figures 2 and 6, showing the cirrus clouds from Figure 6 and the flight path from Figure 2. Please also draw the backward trajectory from Figure 5 in this plot.

**(S 8) Line 184f:** '*In Fig. 8a, the measured Ni is presented and it exceeds the concentration of mineral dust particles or other potential heterogeneous INPs, providing strong evidence that homogeneous freezing played a significant role in shaping the Ni distribution.*'

As you stated earlier in the paper (page 3) 'The limited reliable observation capability of the 2D-S probe above 15 μm restricts obtaining accurate information about young cirrus clouds with high number concentrations of smaller-sized particles…' . Now you use Ni to conclude that homogeneous freezing occurred, correctly arguing that Ni is higher than the concentration of potential heterogeneous INPs. However, I think it should be mentioned again here that Ni does not correspond to the actual ice particle concentration since the small ice particles are missing.

**(S 9) Line 187f:** '*The MMS measured Si in Fig. 8B …*'     Isn't Si from HWV measurements?

**(S 10) Line 189ff:** '*Observing high Si required for homogeneous freezing is inherently challenging, especially within fully developed cirrus clouds, as the available humidity rapidly decreases following a homogeneous freezing event. …*'

What you have written is certainly correct. However, it is possible to find homogeneous events in the measurements, if they are not too old, because they have a signature of high IWCs together with high Si.

I couldn't resist looking at the MacPex 1Hz data from this flight (see plots below) and indeed I would interpret the two events circled in green as homogeneous freezing events – IWC and Si go up to high values and the vertical velocity fluctuations are also quite high. The in-cloud RHi of the younger event around 223K is close to the homogeneous freezing threshold, while the event around 232K appears to be already aged with lower RHi. Outside of the cirrus, RHi is only slightly above saturation, suggesting that cirrus formation probably started with heterogeneous freezing, followed by a subsequent homogeneous freezing event.

[Figure]

**(S 11) Line 200f:** '*The median Ni is about an order of magnitude lower in the lower parts of cirrus which could be explained by following factors:*
*– Homogeneous and heterogeneous ice nucleation produces higher number of ice when the temperatures are lower (Jensen et al., 2013b), leading to higher Ni in the upper parts of cirrus.*'

I think the difference in nucleation rates is not that large in this temperature range… the next points sounds better

'*– The WB-57F collected statistically significant data between 9–11.2 km, covering only the lower portion of the cirrus cloud, where Si is strongly influenced by sedimenting ice crystals. At 10 km, the formation of new ice crystals is more unlikely than at 11.2 km, as most ice at this altitude likely originated from higher layers. Competition for available water vapor further reduces the potential for ice nucleation.*'

**(S 12) Line 251ff:** '*Ahola et al. (2020) implemented various freezing mechanisms, … . By default, UCLALES-SALSA uses homogeneous and heterogeneous ice nucleation parametrization schemes based on Khvorostyanov and Curry (2000), however, due to the heterogeneous ice nucleation schemes being stochastic (time dependent), it was concluded that ice nucleation could be greatly over-estimated in this particular study. To overcome this issue, a deterministic, time-independent deposition nucleation parametrization developed by Ullrich et al. (2017) for uncoated mineral dust particles was implemented to SALSA. The reparametrization was created by using a fit to ice nucleation activity of several mineral dust particles presented in Kanji et al. (2011). For this scheme, tracking of activated INP fractions is necessary since deterministic parametrizations base their ice nucleation activity on original INP population. Homogeneous freezing is implemented based on the temperature and Si relation presented in (Koop et al., 2000).*'

**a)** I would move this paragraph to Section 2.3 UCLALES-SALSA, the information is missing there, see my point **(S 5).**

**b)** One could also consider moving the entire Section 4 to Section 2.3.

**(S 13) Line 274f:** '*These runs are referred to as STND (standard) and AGED respectively from hereafter.*'

Please provide a table in which the conditions of all simulation set ups (STND, AGED, ADJ, HOM) are summarized.

**(S 14) Figure 12:** Distributions of simulated and observed Ni are shown.

**a)** An important point not mentioned in the paper is the size range over which the simulated Ni is calculated. Only the 2D-S size interval (> 15 um) should be considered. Otherwise the simulated Ni are not comparable with the measurements.

If smaller ice crystals are included in the simulated Ni, the analyses should be repeated for the appropriate size interval.

**b)** Figure 12 is not discussed until after Figure 13 - I recommend that it is only shown after Figure 13, as it sums up the results of the study.

**(S 15) Line 409ff:** *'Among all cases in this study, the HOM simulations show the closest statistical resemblance to the 2DS measurements, proving that without any influence of heterogeneous ice nucleation the Ni exceed the clear air concentration of mineral dust most efficiently.'*

This is true for the upper layer of the cirrus cloud, but not for the lower, where the AGED fits best to the measurements. This is not clear from the text.

As this part of the manuscript is of great importance, I recommend that the text be revised accordingly.

**(S 16) Figure 14, Section 5.4 :** Frequency distributions of vertical wind.

I strongly recommend to show this this Figure earlier. While reading the discussion of Figs. 11 and 13, I have been wondering the whole time how the fluctuations of the vertical wind in the model correspond to the measured ones.

It would fit in in Section 4, or, as recommended in S 12 / S 6, all the relevant information on the simulations in Section 2.3?

**(S 17) Line 435-439**: *'It was stated previously that the maximum Ni achieved in HOM cases was not clearly correlated to the imposed large-scale w. The cooling within the supersaturated layer was primarily influenced by the large-scale w; however, small-scale turbulence induced local variations in temperature and humidity. …. '*
The whole paragraph would be better included in the discussion of Figs. 11 and 13.

**(S 18) Section 5.4.1:** This section (including Fig.15) would also be better included in the discussion of Figs. 11 and 13.

**(S 19) Figure 15:**   For better comparison, please synchronise the y-axes (temperature and Ni) of panels (a) and (b).

**(S 20) Line 482f:** *'…. the cirrus clouds on April 16th, 2011, were predominantly formed through homogeneous freezing, …'*

…    the top layer of  cirrus clouds on April 16th, 2011, were predominantly formed through homogeneous freezing,   …

 **(S 21) Line 486ff:**  ' *Simulations with measured mineral dust concentrations (STND) showed an almost complete absence of homogeneous freezing. This suggests that prior heterogeneous nucleation events likely depleted the heterogeneous INPs from certain layers of the cirrus clouds, particularly in the colder upper regions.'*

Something is weird  here ...  why does complete absence of homogeneous freezing  suggest that prior heterogeneous nucleation events likely depleted the heterogeneous INPs?

**(S 22)**  Did I miss it or is there no information about the heterogeneous freezing threshold?

---

## Author Response (AR1)

**Prior heterogeneous ice nucleation events increase likelihood of homogeneous freezing during the evolution of synoptic cirrus – Responses to RCs**

Kasper Juurikkala, Christina J. Williamson, Karl D. Froyd, Jonathan Dean-Day, Ari Laaksonen 29th of April 2025

We thank the editor for handling the review of our work. Our detailed responses to the RCs are provided in the following sections.

**1 Response on RC1**

**General comment:**

This manuscript examines the formation mechanism of a synoptic cirrus observed during the MACPEX field campaign. The observations suggest homogeneous freezing as the dominant nucleation mechanism, which is investigated through simulations with the UCLALES-SALSA model. Generally, the manuscript deals with the topic of the influence of heterogeneously freezing particles (so-called INPs, mineral dust in this case) on the homogeneous freezing process. The theory of the microphysics of this process is well understood and the study does not provide new microphysical insights. What is presented are sensitivity studies with a model in which the measured properties of a cirrus, namely that it is formed by pure homogeneous freezing, can be confirmed by theoretical considerations. That means in summary the agreement between theory and observations is shown. Specifically, the study shows that pure homogeneous freezing took place in the upper layers of the observed cirrus, because the previously existing INPs had already frozen out earlier. The study also examines in general the influence of INPs on the formation process -heterogeneous or homogeneous - of cirrus clouds in the specific dynamic situation of the observed cirrus clouds. The influence of small-scale fluctuations of vertical velocity is also discussed. Overall, the study is interesting and timely and the methods used are appropriate, making the manuscript suitable for publication in ACP.

**General comment 1:**

However, I find the title a bit exaggerated and not quite appropriate for the study, even though I understand that the title is intended to arouse curiosity. I suggest toning down the title to something like 'Prior heterogeneous ice nucleation events shape homogeneous freezing during the evolution of synoptic cirrus' or 'Prior heterogeneous ice nucleation events enable / allow pure homogeneous freezing during the evolution of synoptic cirrus' or, completely different: 'The influence of heterogeneous ice nucleation on cirrus evolution: a case study of an observed homogeneously formed cirrus' which to my feeling better reflect the content of the study. This content could/should also be better elaborated in the paper.

**General comment 2:**

In general, I have to say that the study could/should be significantly improved in terms of writing. In its current form, the reader cannot follow the content fluently - it took me a long time to put the puzzle together, and if I hadn't had to review the paper, I certainly wouldn't have read it to the end....

Often (a) references to previous work is missing; also,

(b) the model description is too brief. For more detail see the see specific comments.

Further, (c) the manuscript is not well structured, which makes it unnecessarily difficult to understand. As outlined in the specific comments, I recommend to

- distribute Section 5.4 to Section 4 and Sections 5.2/5.3 and
- move Fig. 12 behind Figs. 11 and 13.

This would give the manuscript a clearer structure and Fig. 12 - the summary of the results would be the final figure to the paper, which I think is a better place then now in between more detailed results. A further idea might be to then – merge sections 4 and 2.3 to have only one section describing the model and the simulated scenarios. My final assessment is that the paper needs fundamental revision, in terms of the focus and findings of the study, the introduction to the topic, the incorporating of previous work and a more fluent structure.

**Response**: We first would like to thank the referee for the thorough and nice review of our manuscript with the detailed comments and questions, which will be addressed point by point below.

**Reponse to G1**: It is true that the title sounds a little bit exaggerated as the study was limited to a single cloud study. Thus we have decided to go with the title suggestion 'Prior heterogeneous ice nucleation events shape homogeneous freezing during the evolution of synoptic cirrus'. This title resonates better with the narrative presented in the original manuscript.

**Response to G2**: Thank you for providing additional relevant references (point a). Additional background to modeling has been added to the introduction along with references and suggested references from RC2. Point b has been addressed further in the response to the comment 5. I have a short summary of all of the major structural changes in the following:

Section 3.1 was combined and shortened as a single Section 3 (Case study-16 April 2011). In addition, Fig. 3 was removed due to overlap of information with new Fig. 2 (response Fig. 2). Parts of model setup (Section 4) was moved to SI. The Section 5.4 (the effects of atmospheric fluctuations) was moved after Section 4.1 (the effect of aged mineral dust on ice nucleation activity) and the initial part was incorporated to new Section 3.1 (indication of homogeneous freezing) included in the discussion of indication of homogeneous freezing in the measurements. The discussion in the Section 5.4.1 was removed. The Section 3.3 was removed and parts of the text were incorporated as a part of indication of homogeneous freezing discussion (Sect. 3.1). Also, a part of paragraph presenting the concentration of mineral dust and other aerosols was moved to the new Sect 3.1. With these modifications to the structure, the overall manuscript has a greatly improved readability.

**Specific comments:**

Comment 1: Line 33f: 'Over the past few decades, several key measurement campaigns (e.g., Krämer et al., 2009; Voigt et al., 2017) have been conducted in the UTLS.'

Please add more recent work here:

- Krämer et al., 2009 reported multiple campaigns, the following studies could be added:

Krämer et al., 2016 (ACP), Krämer et al. 2020 (ACP), Patnaude et al., 2021 (ACP), Ngo et al., 2024 (ACP). Voigt et al. (2017) presents a single field campaign (ML-Cirrus), the following studies could be added:

i.e. Pan et al. (2010) (START08, BAMS), Wendisch et al. (2016) (ACRIDICON-CHUVA, BAMS), Jensen et al. (2017) (ATTREX, BAMS), Pan et al. (2017) (CONTRAST, BAMS). These campaigns are included either in Krämer et al. (2020) or in Ngo et al. (2024) (or both).

Response: Suggested campaign studies are added to the list of references as follows.

Lines 33-35 (in the revised manuscript): 'Over the past few decades, several key measurement campaigns Added references e.g., Pan et al. (2010, START08,BAMS), Jensen et al. (2013b, MACPEX), Wendisch et al. (2016, ACRIDICON-CHUVA), Jensen et al. (2017, ATTREX,BAMS), Pan et al. (2017, CONSTRAST) have been conducted in the UTLS.'

**Comment 2**: Line 33: 'Synoptic cirrus clouds primarily form through two dominant mechanisms: heterogeneous and homogeneous freezing.'

This applies not only to synoptic, but to all cirrus.

**Response**: We have rewritten that sentence to clarify our statement:

Lines 39-40: 'Cirrus clouds primarily form through two dominant mechanisms: heterogeneous and homogeneous freezing (Pruppacher and Klett, 1997).'

Comment 3: Line 38ff: 'Heterogeneous ice nucleation . . . In contrast, homogeneous freezing . . . '

Please provide references for heterogeneous and homogeneous freezing.

**Response**: Reference added as in the response to comment 2.

**Comment 4**: Line 48ff: 'Ice nucleation in cirrus clouds is strongly influenced by the abundance of INPs, which regulate how efficiently heterogeneous ice nucleation can suppress homogeneous freezing and activity.'

Please provide references for the competition between heterogeneous and homogeneous freezing, i.e. influence of INP number and vertical velocity on the onset of homogeneous freezing. however, the present study lacks references to previous work. Here are some examples, but the list is not exhaustive.

- Spichtinger, P., and D. J. Cziczo (2010), Impact of heterogeneous ice nuclei on homogeneous freezing events in cirrus clouds, J. Geophys. Res., 115, D14208, doi:10.1029/2009JD012168.
- Rolf, C., Krämer, M., Schiller, C., Hildebrandt, M., and Riese, M.: Lidar observation and model simulation of a volcanic-ash-induced cirrus cloud during the Eyjafjallajökull eruption, Atmos. Chem. Phys., 12, 10281–10294, https://doi.org/10.5194/acp-12-10281-2012, 2012.
- Krämer, M., Rolf, C., Luebke, A., Afchine, A., Spelten, N., Costa, A., Meyer, J., Zöger, M., Smith, J., Herman, R. L., Buchholz, B., Ebert, V., Baumgardner, D., Borrmann, S., Klingebiel, M., and Avallone, L.: A microphysics guide to cirrus clouds Part 1: Cirrus types, Atmos. Chem. Phys., 16, 3463–3483, https://doi.org/10.5194/acp-16-3463-2016, 2016.
- Kärcher, B., Jensen, E. J., Lohmann, U. (2019). The impact of mesoscale gravity waves on homogeneous ice nucleation in cirrus clouds. Geophysical Research Letters, 46, 5556–5565. https://doi.org/10.1029/2019GL082437

The authors miss to discuss this paper, although it also compares MACPEX observations with simulations.

- Kärcher, B., DeMott, P. J., Jensen, E. J., Harrington, J. Y. (2022). Studies on the competition

between homogeneous and heterogeneous ice nucleation in cirrus formation. Journal of Geophysical Research: Atmospheres, 127, e2021JD035805. https://doi.org/10.1029/2021JD035805

**Response**: Introduction to advances in modeling studies added with appropriate references as following:

Ice nucleation in cirrus clouds is shaped by the abundance of ice-nucleating particles (INPs), which govern the efficiency with which heterogeneous ice nucleation suppresses homogeneous freezing. In heterogeneous nucleation, INPs trigger ice formation and are subsequently removed from higher altitudes as ice crystals sediment, whereas homogeneous freezing, occurring without INPs, produces ice crystal concentrations that vary with environmental factors such as temperature and vertical velocity (Kärcher and Lohmann, 2002). Modeling studies have significantly advanced our grasp of these processes (e.g. Sassen and Benson, 2000; Kärcher and Lohmann, 2002; Spichtinger and Cziczo, 2010; Rolf et al., 2012; Krämer et al., 2016; Kärcher et al., 2019, 2022). Efforts span a range of scales, from high-resolution Large Eddy Simulations (Sölch and Kärcher, 2010) to global models (Liu et al., 2012; Tully et al., 2022; Beer et al., 2024). Yet, despite these advances, accurately representing INPs remains challenging due to their scarcity (0.01 to 100 L-1 at -30°C; DeMott et al. (2010)) and varied sources, underscoring the need for refined parameterizations and robust observational constraints (Burrows et al., 2022).

**Comment 5: Section 2.3 UCLALES-SALSA**

- a) The information about the model is very limited. I recommend a short description of which parameterizations for heterogeneous and homogeneous freezing are used and how the processes (ice particle growth, evaporation, sedimentation, aggregation, ...) are treated in the model.
- b) Further, please provide information about the sizes of the ice bins

**Response**: The following part from Section 4 was moved to Section 2.3 to clarify this issue:

'Ahola et al. (2020) implemented various freezing mechanisms, however, in this study, mainly homogeneous freezing and deposition nucleation are turned on as the typical temperatures within upper tropospheric cirrus clouds are below -38° where pure water does not stay in liquid state. By default, UCLALES-SALSA uses homogeneous and heterogeneous ice nucleation parametrization schemes based on Khvorostyanov and Curry (2000), however, due to the heterogeneous ice nucleation schemes being stochastic (time dependent), it was concluded that ice nucleation could be greatly over-estimated in this particular study. To overcome this issue, a deterministic, time-independent deposition nucleation parametrization developed by Ullrich et al. (2017) for uncoated mineral dust particles was implemented to SALSA. The reparametrization was created by using a fit to ice nucleation activity of several mineral dust particles presented in Kanji et al. (2011). For this scheme, tracking of activated INP fractions is necessary since deterministic parametrizations base their ice nucleation activity on original INP population. Homogeneous freezing is implemented based on the temperature and  $S_i$  relation presented in Koop et al. (2000).

Once ice nucleation occurs, ice crystals are transferred from aerosol bins to ice bins, where their subsequent evolution is governed by microphysical processes such as vapor deposition, sublimation, and sedimentation. The growth and evaporation of ice crystals are calculated using the framework from Jacobson (2005). Ice sedimentation is explicitly resolved based on size-dependent terminal velocities. Additionally, ice-ice collisions are aggregation, influencing the size distribution of ice crystals over time.

Figure 1: (a) Distribution of  $N_i$  measured with 2DS inside cirrus clouds at two constant altitude levels with most continuous time series measured. Dashed line is drawn at the concentration of the mean clear-air concentration of mineral dust  $(2.05\times10^{-2}~{\rm cm}^{-3})$  by PALMS+FCAS. (b) Supersaturation over ice measured at two constant altitude levels. Blue shading indicates homogeneous freezing threshold  $S_i$  between 210-230 K based on equation presented in Ren and Mackenzie (2005). (c) Ice water content (IWC) is from CLH measurements. Sampling rate of each instrument was at 1 Hz.

To represent ice crystal size distributions, the ice bins in SALSA range from 2 to 400  $\mu$ m in diameter, ensuring a comprehensive resolution of both small ice crystals and larger particles. Ice crystals below 40  $\mu$ m are assumed to be spherical, while those above this threshold are treated as bullet rosettes, following the habit-dependent parameterization in Baum et al. (2005). The habit transition is consistent with in-situ cirrus observations that show a prevalence of bullet rosettes in larger particle size ranges.'

**Comment 6**: Figure 4: What time resolution do the measurements shown have? Also, why not showing the IWC (ice water content) in addition? There was a total water instrument on board (CLH), so together with the HVW gas phase water you have a good IWC information.

**Response**: The sampling rate of each instrument shown in response Fig. 1 is 1 Hz. Instead of adding a vertical distribution of IWC and  $N_i$  to the 'then' Fig. 4, a frequency distribution of IWC is added to new Fig. 6 as in response the Fig. 1:

**Comment 7: Figures 2, 5, 6:**

To give the reader a better overview, I recommend making one plot from Figures 2 and 6, showing the cirrus clouds from Figure 6 and the flight path from Figure 2. Please also draw the backward trajectory from Figure 5 in this plot

**Response:** The changed Figure shown in response Fig.2:

**Comment 8: Line 184f:**

'In Fig. 8a, the measured Ni is presented and it exceeds the concentration of mineral dust particles or other potential heterogeneous INPs, providing strong evidence that homogeneous freezing played a significant role in shaping the Ni distribution.' As you stated earlier in the paper (page 3) 'The limited reliable observation capability of the 2D-S probe above 15  $\mu$ m restricts obtaining accurate information about young cirrus clouds with high number concentrations of smaller-sized particles...'. Now you use Ni to conclude that homogeneous freezing occurred, correctly arguing that Ni is higher than the concentration of potential heterogeneous INPs. However, I think it should be mentioned

Figure 2: GOES-16 satellite imagery shows infrared radiation (IR) temperature of cirrus cloud tops for 8.15 UTC (top-left focused panel) and 18.15 UTC (base). A back-trace trajectory trajectory with black dashed line is shown, starting from the aircraft's intersection with the cirrus clouds at 18 UTC. The 8.15 UTC imagery with effective cloud-top temperatures below -40°C indicates intense cirrus clouds. A blue cross is marked in the centre of this panel to indicate where the air parcel was located around 8.15 UTC in the back-trace trajectory. The flight path of the WB-57F on April 16th, 2011, is overlaid with coloured lines representing the aircraft's position. The colour of the flight path corresponds to altitude and time in UTC.

again here that Ni does not correspond to the actual ice particle concentration since the small ice particles are missing.

**Response:** We adjusted the  $N_i$  distributions to reflect the ice crystals over 15  $\mu$ m (response Fig. 3). The adjustment had no influence on the conclusions of the study due to limited effect on the distribution shape.

**Comment 9: Line 187f:**

'The MMS measured Si in Fig. 8B . . . ' Isn't Si from HWV measurements?

**Response:** Indeed, the Si is a product of HWV with combination of temperature from MMS. The text is edited with following change

Line 208: 'The HWV-MMS measured  $S_i$  in Fig...'

**Comment 10: Line 189ff:**

'Observing high Si required for homogeneous freezing is inherently challenging, especially within fully developed cirrus clouds, as the available humidity rapidly decreases following a homogeneous freezing event. . . . . '

What you have written is certainly correct. However, it is possible to find homogeneous events in the measurements, if they are not too old, because they have a signature of high IWCs together with high Si. I couldn't resist looking at the MacPex 1Hz data from this flight (see plots below) and indeed I would interpret the two events circled in green as homogeneous freezing events – IWC and

Si go up to high values and the vertical velocity fluctuations are also quite high. The in-cloud RHi of the younger event around 223K is close to the homogeneous freezing threshold, while the event around 232K appears to be already aged with lower RHi. Outside of the cirrus, RHi is only slightly above saturation, suggesting that cirrus formation probably started with heterogeneous freezing, followed by a subsequent homogeneous freezing event.

**Response:** Thank you for your interest in looking deeper into the data! Your observation definitely does sugget that high IWC has a connection to homogeneous freezing. We have added the IWC distribution to the new Fig. 6 (response Fig. 1, and we state that the homogeneous freezing assumption is supported by the high IWC associated with high  $S_i$  in the following form:

'Nevertheless, the high levels of ice water content (IWC) observed within the cirrus clouds (Fig. 6c), occasionally coinciding with elevated  $S_i$  values—indicative of recent homogeneous freezing events (see supplementary Fig. S5)—support the hypothesis that these cirrus clouds most likely formed via homogeneous freezing (Krämer et al., 2016).'

**Comment 11: Line 200f:**

'The median Ni is about an order of magnitude lower in the lower parts of cirrus which could be explained by following factors: – Homogeneous and heterogeneous ice nucleation produces higher number of ice when the temperatures are lower (Jensen et al., 2013b), leading to higher Ni in the upper parts of cirrus.'

I think the difference in nucleation rates is not that large in this temperature range... the next points sounds better

'- The WB-57F collected statistically significant data between 9–11.2 km, covering only the lower portion of the cirrus cloud, where Si is strongly influenced by sedimenting ice crystals. At 10 km, the formation of new ice crystals is more unlikely than at 11.2 km, as most ice at this altitude likely originated from higher layers. Competition for available water vapor further reduces the potential for ice nucleation.'

**Response:** The ice nucleation activity of mineral dust particles, as represented by the Ullrich et al. (2017) parameterization, exhibits relatively large differences at these temperatures. However, as you correctly pointed out, this parameterization may not perfectly capture real-world ice nucleation behavior. Given the associated uncertainties, the actual significance of this activity difference remains unclear. To ensure the robustness of our conclusions, we have removed this point, as its contribution is highly uncertain.

**Comment 12: Line 251ff:**

'Ahola et al. (2020) implemented various freezing mechanisms, . . . . By default, UCLALES-SALSA uses homogeneous and heterogeneous ice nucleation parametrization schemes based on Khvorostyanov and Curry (2000), however, due to the heterogeneous ice nucleation schemes being stochastic (time dependent), it was concluded that ice nucleation could be greatly over-estimated in this particular study. To overcome this issue, a deterministic, time-independent deposition nucleation parametrization developed by Ullrich et al. (2017) for uncoated mineral dust particles was implemented to SALSA. The parametrization was created by using a fit to ice nucleation activity of several mineral dust particles presented in Kanji et al. (2011). For this scheme, tracking of activated INP fractions is necessary since deterministic parametrizations base their ice nucleation activity on original INP population. Homogeneous freezing is implemented based on the temperature and Si relation presented in (Koop et al., 2000).'

- a) I would move this paragraph to Section 2.3 UCLALES-SALSA, the information is missing there, see my point (S 5).
- b) One could also consider moving the entire Section 4 to Section 2.3.

**Response:** This paragraph was moved to section 2.3 to clarify the missing information on ice nucleation parameterizations. We keep the model description and the model setup as separate sections as it is important in our opinion to present the model first, then the background of the study and then proceed to the model set up with the information from the background.

**Comment 13: Line 274f:**

'These runs are referred to as STND (standard) and AGED respectively from hereafter.' Please provide a table in which the conditions of all simulation set ups (STND, AGED, ADJ, HOM) are summarized.

**Response:** Following summarizing table is added to the revised manuscript.

Table 1: Dust profile, ice nucleation mechanisms and humidity profile used in model run setups.

| Setup | Dust profile                                 | Ice nucleation mechanisms | Humidity profile |
|-------|----------------------------------------------|---------------------------|------------------|
| STND  | const. $2.05 \times 10^{-2} \text{ cm}^{-3}$ | het and hom               | $S_i = 1.05$     |
| AGED  | const. $2.05 \times 10^{-3} \text{ cm}^{-3}$ | het and hom               | $S_i = 1.05$     |
| ADJ   | adjusted profile                             | het and hom               | adjusted profile |
| HOM   | adjusted profile                             | hom                       | adjusted profile |

Comment 14: Figure 12: Distributions of simulated and observed Ni are shown

- a) An important point not mentioned in the paper is the size range over which the simulated Ni is calculated. Only the 2D-S size interval (> 15 um) should be considered. Otherwise the simulated Ni are not comparable with the measurements. If smaller ice crystals are included in the simulated Ni, the analyses should be repeated for the appropriate size interval.
- b) Figure 12 is not discussed until after Figure 13 I recommend that it is only shown after Figure 13, as it sums up the results of the study.

**Response:** a) When the  $N_i$  is restricted to above 15 um, the frequency of higher end concentrations decrease by very limited amount as shown in response Fig.3.

b) Fig. 12 (now Fig. 11) will be kept at the original location as some of the results in the Figure are discussed before 'then' Fig. 13.

**Comment 15: Line 409ff:**

'Among all cases in this study, the HOM simulations show the closest statistical resemblance to the 2DS measurements, proving that without any influence of heterogeneous ice nucleation the Ni exceed the clear air concentration of mineral dust most efficiently.' This is true for the upper layer of the cirrus cloud, but not for the lower, where the AGED fits best to the measurements. This is not clear from the text. As this part of the manuscript is of great importance, I recommend that the text be revised accordingly.

**Response:** As you correctly pointed out, the AGED runs exhibit a better match with observations in the lower parts of the cirrus cloud. This is primarily because, in the AGED simulations, the  $S_i$  reaches the threshold for homogeneous freezing more easily in the lower cirrus. This behavior is due to the higher initial  $S_i$  in the standard humidity profile used in these runs.

In contrast, the HOM runs-based on the adjusted humidity profile-start with a lower  $S_i$ , which makes it more difficult for the lower parts of the cloud to reach the levels required for homogeneous nucleation. As a result, these runs show fewer high  $N_i$  events. Additionally, in the HOM setup, ice nucleation is delayed, allowing more time for sedimentation of ice crystals from the upper parts of the

Figure 3: Distribution of  $N_i$  in two altitudes that the WB-57F measured continuous statistical data. The measurement data is from 2DS instrument which is in solid black lines. The STND and AGED runs stand for standard with measured mineral dust concentration, AGED for mineral dust particles that have aged. ADJ stands for adjusted profile set up runs and HOM for homogeneous freezing only runs. PALMS measured mineral dust concentration shown with black dashed vertical line at  $2.05 \times 10^{-2}$  cm-3.

cloud to influence the lower layers, further suppressing local nucleation. To clarify this distinction, we have modified and added new parts to the following explanation to the manuscript:

'Among all the presented simulations, the HOM runs exhibit the closest statistical agreement with the 2DS measurements. This suggests that in the absence of heterogeneous nucleation, homogeneous freezing can generate  $N_i$  values that most effectively exceed the background concentration of mineral dust. In the 9.5–10.5 km range, the AGED simulation matches the observed  $N_i$  distribution most closely. This agreement is primarily due to the initially high and uniform  $S_i$  profile and the relatively low abundance of heterogeneous ice-nucleating particles in that case. However, considering the evolution of humidity following multiple nucleation events, a uniformly high  $S_i$  profile at the start of the model runs is likely an unrealistic representation of real atmospheric conditions. Such profiles would typically require vertically differential cooling rates and very small  $N_i$  that would not remove ice supersaturation.'

**Comment 16: Figure 14, Section 5.4:**

Frequency distributions of vertical wind. I strongly recommend to show this this Figure earlier. While reading the discussion of Figs. 11 and 13, I have been wondering the whole time how the fluctuations of the vertical wind in the model correspond to the measured ones. It would fit in in Section 4, or, as recommended in S 12 / S 6, all the relevant information on the simulations in Section 2.3?

**Response:** An introduction to atmospheric fluctuations was added as a part of discussion in new Sect. 3.1 with the old Fig. 11. Also, the original Section includes overalapping information on the relationship between homogeneous freezing and vertical winds and thus was shortened in the following form:

**'Part of 3.1 Indication of homogeneous freezing**

Finally, the vertical winds measured within the cirrus were of significantly higher magnitude than

Figure 4: (a) Standard deviation of vertical wind and (b) relative standard deviation of supersaturation over ice  $S_i$  shown as shading inside a model domain at cirrus forming altitudes. The velocity fluctuations are generally the same for every set up of simulations used in this study. The variability for  $S_i$  was higher at the end of the model runs due to the ice nucleation and subsequent vapor growth of ice crystals affecting the distribution of  $S_i$  inside the model domain.

those typically associated with synoptic-scale motions, as shown in Fig. 6. This suggests that  $S_i$  may have experienced considerable short-timescale variability, enabling the threshold for homogeneous freezing to be reached through strong upward motions. Figure 6 presents the frequency distributions of observed vertical wind velocities during constant-altitude flight legs, with the vertical motions caused by aircraft ascents and descents and short scale systematic error filtered out.'

**Comment 17: Line 435-439:**

'It was stated previously that the maximum Ni achieved in HOM cases was not clearly correlated to the imposed large-scale w. The cooling within the supersaturated layer was primarily influenced by the large-scale w; however, small-scale turbulence induced local variations in temperature and humidity. . . . . 'The whole paragraph would be better included in the discussion of Figs. 11 and 13.

**Response:** This discussion was moved as a part of the new Section 4.3 (Accounting for prior cirrus formation).

**Comment 18: Section 5.4.1:**

This section (including Fig.15) would also be better included in the discussion of Figs. 11 and 13.

**Response:** This section was removed as the results presented provide little to the article conclusions and to respond to the RC2 comments on reducing the overall length of the manuscript.

**Comment 19**: Figure 15: For better comparison, please synchronise the y-axes (temperature and Ni) of panels (a) and (b).

**Response:** This Figure was replaced by another response Fig. 4 showing the variation in both vertical wind and  $S_i$  in more broader term. The removal was done because the Sect. 5.4.1 was removed.

**Comment 20: Line 482f:**

".... the cirrus clouds on April 16th, 2011, were predominantly formed through homogeneous freezing, ..." ... the top layer of cirrus clouds on April 16th, 2011, were predominantly formed through homogeneous freezing, ...

Response: Edited as suggested.

**Comment 21: Line 486ff:**

'Simulations with measured mineral dust concentrations (STND) showed an almost complete absence of homogeneous freezing. This suggests that prior heterogeneous nucleation events likely depleted the heterogeneous INPs from certain layers of the cirrus clouds, particularly in the colder upper regions.' Something is weird here ... why does complete absence of homogeneous freezing suggest that prior heterogeneous nucleation events likely depleted the heterogeneous INPs?

Response: The heterogeneous ice formed due to the number of heterogeneous INPs measured was preventing Si from reaching high enough for homogeneous freezing in the simulations. A clarification was added to the leading sentence in the following form: 'Simulations with measured mineral dust concentrations (STND) showed an almost complete absence of homogeneous freezing due to heterogeneous ice preventing supersaturation over ice from reaching the critical threshold level for homogeneous freezing.'

Comment 22: Did I miss it or is there no information about the heterogeneous freezing threshold?

**Response:** A clear heterogeneous freezing threshold does not exist but generally the activity of mineral dust particles follows the Ullrich et al. (2017) parameterization. Information on frozen fraction given by the deposition nucleation parameterization in supplements.

**2 Response on RC2**

**General comment:**

In this paper Juurikkala et al. use a large-eddy model to show that early heterogeneous ice nucleation can lead to a depletion of ice nucleating particles and later instances of homogeneous freezing as a result. Data from a NASA study, MACPEX, were used for this work. This is largely in agreement with other recent field studies. It is noteworthy that the model result – showing that initial heterogeneous freezing can lead to later homogeneous freezing – has been previously shown in a number of other model studies (as referenced here, e.g., Spichtinger et al., etc.). None the less, this is an important addition to the literature and one that should be published in ACP given some minor changes.

My major comment is that the overall length of the paper, 30 pages, is far in excess of what is needed for the treatment of what is essentially a single cloud case study. This is specifically true of section 3, the case study (6 pages), 4, the model setup (4 pages) and 5, the comparison of model and observations (9 pages). Since this paper contains supplementary information, I would highly encourage the authors to move as much of this as possible to the SI. I believe the text in these sections could easily be reduced by half without loss of content. The tables and half the figures (in section 3 Figures 2 overlaps greatly with 3 and 6, ideally just retain 3 and move the others to SI, in the later sections all contain 2 panels of which 1 can easily move to SI) could also be moved to SI. The use of so many sub-sections which seem to be small scale breaks in content also makes what should be a short and focused paper seem very chopped up. My recommendation is also that comparison to the field studies and the model description could largely rely on published results without the need for the repetition here. I hope the authors consider this as the paper will lose a lot of readers in the great length. Pulling out and focusing on the novel material – by eliminating or moving to SI the non-critical content - would increase readability a great deal.

I also encourage the authors to clarify from the abstract and introduction that the paper is a single cloud case from MACPEX, as treated by Jensen et al. 2013(b in the reference list), not an overview of the whole field mission. The ice residual measurements described in Cziczo et al., 2013 (also referenced), for the whole mission, indicate the prevalence of supersaturations and ice residuals consistent with heterogeneous freezing (not in this case but overall). This distinction is unfortunately not clearly stated until the Conclusions and so appears contradictory unless one reads all the reference materials in detail; the paper would therefore benefit from a statement in the abstract and a short paragraph in the introduction. I believe this would both help motivate and clarify their paper.

Another missing element is a couple paragraphs of treatment of previous modeling studies in the introduction, specifically those that considered the interplay between homogeneous and heterogeneous freezing. The absence of description of previous work – which, conversely, is done for field measurements - leads one to believe this is the first time it has been done, which is clearly not the case. There are some statements that discuss comparison to other studies but they are imbedded in the Case Study and Model Setup sections. Ideally these should be gathered to the Introduction, as is done for comparisons of these data to previous field measurements. Therefore, inclusion of a paragraph or two on previous work should be straightforward.

Overall, this paper, after significant shortening, will be a solid addition to the literature on the contemporary important topic of ice nucleation m

Response: We thank the referee for the input for the improvement suggestions. A summary of

the general modifications to the manuscript is as follows: As the RC1 suggested, Figs. 2 and 6 were combined to decrease the amount of overlapping information. Section 3.1 was combined and shortened as a single Section 3 (Case study-16 April 2011). In addition, Fig. 3 was removed due to overlap of information with new Fig. 2 (response Fig. 2). Parts of model setup was moved to SI. The Section 5.4 (the effects of atmospheric fluctuations) was moved after Section 4.1 (the effect of aged mineral dust on ice nucleation activity) and the initial part was incorporated to new Section 3.1 (indication of homogeneous freezing) included in the discussion of indication of homogeneous freezing in the measurements. The discussion in the Section 5.4.1 was removed. The Section 3.3 was removed and parts of the text were incorporated as a part of indication of homogeneous freezing discussion (Sect. 3.1). Also, a part of paragraph presenting the concentration of mineral dust and other aerosols was moved to the new Sect 3.1. With these modifications to the structure, the overall manuscript has a greatly improved readability.

Also, we understand your opinion that the study seems to be excessively long for a single cloud case study. An effort was made to reduce the length of the introduction and discussion and the length of the complete manuscript was reduced from 30 to 27 pages. As you pointed out, previous modeling studies and clarification of this study being only a single cloud case study was added to the introduction as answered for the RC1 Comment 4. Also, the beginning of the second paragraph in the abstract was changed to this wording: '... In a single cloud case study...'.

---

## Referee Report (RR1)

**Second review of**

**Prior heterogeneous ice nucleation events increase likelihood of homogeneous freezing during the evolution of synoptic cirrus**

Kasper Juurikkala et al.

Previous comments are in black, responses in blue and new comments are in green

**New General:**

The manuscript has improved significantly. The authors invested considerable effort into the study, and their work has paid off. The paper is now nearly ready for publication. However, I still have a few comments -both on previously raised issues and on some new points- which are listed below.

**Specific comments:**

(S 1) Line 33f: 'Over the past few decades, several key measurement campaigns (e.g., Krämer et al., 2009; Voigt et al., 2017) have been conducted in the UTLS.'

Please add more recent work here:

- Krämer et al., 2009 reported multiple campaigns, the following studies could be added: Krämer et al., 2016 (ACP), Krämer et al. 2020 (ACP), Patnaude et al., 2021 (ACP), Ngo et al., 2024 (ACP).

Voigt et al. (2017) presents a single field campaign (ML-Cirrus), the following studies could be added:

i.e. Pan et al. (2010) (START08, BAMS), Wendisch et al. (2016) (ACRIDICON-CHUVA, BAMS), Jensen et al. (2017) (ATTREX, BAMS), Pan et al. (2017) (CONTRAST, BAMS).

These campaigns are included either in Krämer et al. (2020) or in Ngo et al. (2024) (or both).

**Response**: Suggested campaign studies are added to the list of references as follows. Lines 33-35 (in the revised manuscript):

'Over the past few decades, several key measurement campaigns e.g., Pan et al. (2010, START08,BAMS), Jensen et al. (2013b, MACPEX), Wendisch et al. (2016, ACRIDICON-CHUVA), Jensen et al. (2017, ATTREX,BAMS), Pan et al. (2017, CONSTRAST) have been conducted in the UTLS.'

**New comment**: That's not what I meant - now only a few of all the campaigns are mentioned... I suggest

'Over the past few decades, a number of key measurement campaigns have been conducted in the UTLS, which are compiled by Krämer et al. (2016, 2020) and Ngo et al. (2025).'

(S 3) Line 38ff: 'Heterogeneous ice nucleation ... In contrast, homogeneous freezing ...'

Please provide references for heterogeneous and homogeneous freezing.

**Response:** Reference added as in the response to comment 2. '… (*Pruppacher and Klett, 1997*).'

**New comment**: This is a fairly old reference; since then, the understanding of heterogeneous and homogeneous freezing has evolved considerably. I highly recommend citing more recent references here – and/or point to new Section 2.3.

(S 21) Line 486ff (new 460ff): 'Simulations with measured mineral dust concentrations (STND) showed an almost complete absence of homogeneous freezing. This suggests that prior heterogeneous nucleation events likely depleted the heterogeneous INPs from certain layers of the cirrus clouds, particularly in the colder upper regions.'

Something is weird here ... why does complete absence of homogeneous freezing suggest that prior heterogeneous nucleation events likely depleted the heterogeneous INPs?

**Response**: The heterogeneous ice formed due to the number of heterogeneous INPs measured was preventing Si from reaching high enough for homogeneous freezing in the simulations. A clarification was added to the leading sentence in the following form:

'Simulations with measured mineral dust concentrations (STND) showed an almost complete absence of homogeneous freezing due to heterogeneous ice preventing supersaturation over ice from reaching the critical threshold level for homogeneous freezing.'

"... almost complete absence of homogeneous freezing..." still reads strangely in the context, because the message of the manuscript is that the measured ice crystals formed predominantly homogeneously.

If I understand it correctly, the result that 'the measured ice crystals formed predominantly homogeneously' comes from the ADJ simulations and the homogeneous freezing does not occur in the STND runs. The difference is the moisture profiles.

The moisture profiles for ADJ, which are responsible for the homogeneous freezing, can be seen in Figure 10 (see my comment on Figure 10 below). The difference is that in the STND runs Si does not reach the homogeneous freezing threshold, whereas in the ADJ runs it does.

Therefore, it should be clear from the text that the subsequent dynamic situation must be favorable so that homogeneous freezing can take place after the heterogeneous exhaustion of the INPs.

Since this is the core message of the manuscript, here an attempt to rephrase this section as I would understand it better (and hope it is correct:

'We investigated the role of heterogeneous ice nucleation with the UCLALES-SALSA model, and the results showed that prior heterogeneous ice nucleation increases the likelihood of homogeneous freezing during subsequent ice nucleation events. Simulations with measured mineral dust concentrations (STND) demonstrated that heterogeneous ice nucleation events likely depleted the heterogeneous INPs from certain layers of the cirrus clouds. In addition, ice supersaturation is suppressed below the homogeneous freezing threshold, particularly in the colder upper regions, resulting in an almost complete absence of homogeneous ice nucleation. Under dynamic conditions allowing supersaturations to reach the homogeneous freezing threshold, such as in the ADJ scenario, this prior depletion indirectly enabled the occurrence of homogeneous freezing in later stages.

**(N 1 - New comment) Figures 9 and 13:** I preferred the previous versions of the Figures, particularly since the model scenarios (STND and AGED / ADJ and HOM) were listed above the two columns. Now, you have to figure the scenarios out from the caption. Please reinsert the abbreviations of the scenarios to the right of the rows.

**(N 2 - New comment) Figure 10: In the Figure caption it is mentioned**

'... (b) ice saturation profiles from the beginning to the end of the simulation using the STND setup.' I think that the ADJ scenario is meant?

**(N 3 - New comment) Line 377 (of manuscript with changes tracked):**

'Also the homogeneous freezing is known to be mostly insensitive to the concentration or size distribution of available aerosols, ... '

Homogeneous freezing is sensitive to the size distribution of the aerosols present, in particular at cold temperatures and higher updrafts. as shown in Baumgartner et al. (2023).

Baumgartner, M., Rolf, C., Grooß, J.-U., Schneider, J., Schorr, T., Möhler, O., Spichtinger, P., and Krämer, M.: New investigations on homogeneous ice nucleation: the effects of water activity and water saturation formulations, Atmos. Chem. Phys., 22, 65–91, https://doi.org/10.5194/acp-22-65-2022, 2022.

**(N 4 - New comment) Line 555f (of manuscript with changes tracked):**

'This reflects the known sensitivity of homogeneous freezing to the magnitude of vertical velocity.' I suggest to add here 'and also the weakening of homogeneous nucleation events due to prior heterogeneous ice nucleation (Spichtinger and Cziczo, 2010).

---

## Author Response (AR2)

**Prior heterogeneous ice nucleation events increase likelihood of homogeneous freezing during the evolution of synoptic cirrus – Responses to RCs of second review**

Kasper Juurikkala, Christina J. Williamson, Karl D. Froyd, Jonathan Dean-Day, Ari Laaksonen

15th of August 2025

We thank the editor for handling the review of our work. Our detailed responses to the RCs are provided in the following sections.

**1 Response on RC1**

**General comment:**

The manuscript has improved significantly. The authors invested considerable effort into the study, and their work has paid off. The paper is now nearly ready for publication. However, I still have a few comments -both on previously raised issues and on some new points- which are listed below.

Reponse to G1: We thank the reviewer for providing valuable insight and fix suggestions in the second review of the manuscript. We have clarified each comment below.

**Specific comments:**

Comment 1: Line 33f: 'Over the past few decades, several key measurement campaigns (e.g., Krämer et al., 2009; Voigt et al., 2017) have been conducted in the UTLS.'

Please add more recent work here:

- Krämer et al., 2009 reported multiple campaigns, the following studies could be added: Krämer et al., 2016 (ACP), Krämer et al. 2020 (ACP), Patnaude et al., 2021 (ACP), Ngo et al., 2024 (ACP). Voigt et al. (2017) presents a single field campaign (ML-Cirrus), the following studies could be added:

i.e. Pan et al. (2010) (START08, BAMS), Wendisch et al. (2016) (ACRIDICON-CHUVA, BAMS), Jensen et al. (2017) (ATTREX, BAMS), Pan et al. (2017) (CONTRAST, BAMS). These campaigns are included either in Krämer et al. (2020) or in Ngo et al. (2024) (or both).

Response: Suggested campaign studies are added to the list of references as follows.

Lines 33-35 (in the revised manuscript): 'Over the past few decades, several key measurement campaigns Added references e.g., Pan et al. (2010, START08,BAMS), Jensen et al. (2013b, MACPEX), Wendisch et al. (2016, ACRIDICON-CHUVA), Jensen et al. (2017, ATTREX,BAMS), Pan et al. (2017, CONSTRAST) have been conducted in the UTLS.'

**New comment**: That's not what I meant - now only a few of all to campaigns are mentioned... I suggest

'Over the past few decades, a number of key measurement campaigns have been conducted in the UTLS, which are compiled by Krämer et al. (2016, 2020) and Ngo et al. (2025).'

**Answer to the new comment** The text has been edited in the suggested form.

Comment 3: Line 38ff: 'Heterogeneous ice nucleation ... In contrast, homogeneous freezing ...'

Please provide references for heterogeneous and homogeneous freezing.

Response: Reference added as in the response to comment 2. ...(Pruppacher and Klett, 1997)'

**New comment**: This is a fairly old reference; since then, the understanding of heterogeneous and homogeneous freezing has evolved considerably. I highly recommend citing more recent references here – and/or point to new Section 2.3.

Answer to the new comment The text has been edited in the suggested accordingly with followin form: 'Cirrus clouds primarily form through two dominant mechanisms: heterogeneous and homogeneous freezing (Pruppacher and Klett, 1997; Cziczo and Froyd, 2014; Kanji et al., 2017).' Also references were added to the definition of homogeneous freezing: 'In contrast homogeneous freezing occurs in the absence of INPs and takes place when aqueous solution droplets freeze at temperatures below the -38C threshold for pure water and at high Si (e.g., Schneider et al., 2021; Koop et al., 2000).

**Comment 21: Line 486ff:**

'Simulations with measured mineral dust concentrations (STND) showed an almost complete absence of homogeneous freezing. This suggests that prior heterogeneous nucleation events likely depleted the heterogeneous INPs from certain layers of the cirrus clouds, particularly in the colder upper regions.' Something is weird here ... why does complete absence of homogeneous freezing suggest that prior heterogeneous nucleation events likely depleted the heterogeneous INPs?

Response: The heterogeneous ice formed due to the number of heterogeneous INPs measured was preventing Si from reaching high enough for homogeneous freezing in the simulations. A clarification was added to the leading sentence in the following form: 'Simulations with measured mineral dust concentrations (STND) showed an almost complete absence of homogeneous freezing due to heterogeneous ice preventing supersaturation over ice from reaching the critical threshold level for homogeneous freezing.'

'... almost complete absence of homogeneous freezing...' still reads strangely in the context, because the message of the manuscript is that the measured ice crystals formed predominantly homogeneously.

If I understand it correctly, the result that 'the measured ice crystals formed predominantly homogeneously' comes from the ADJ simulations and the homogeneous freezing does not occur in the STND runs. The difference is the moisture profiles. The moisture profiles for ADJ, which are responsible for the homogeneous freezing, can be seen in Figure 10 (see my comment on Figure 10 below). The difference is that in the STND runs Si does not reach the homogeneous freezing threshold, whereas in the ADJ runs it does. Therefore, it should be clear from the text that the subsequent dynamic situation must be favorable so that homogeneous freezing can take place after the heterogeneous exhaustion of the INPs. Since this is the core message of the manuscript, here an attempt to rephrase

this section as I would understand it better (and hope it is correct: 'We investigated the role of heterogeneous ice nucleation with the UCLALES-SALSA model, and the results showed that prior heterogeneous ice nucleation increases the likelihood of homogeneous freezing during subsequent ice nucleation events. Simulations with measured mineral dust concentrations (STND) demonstrated that heterogeneous ice nucleation events likely depleted the heterogeneous INPs from certain layers of the cirrus clouds. In addition, ice supersaturation is suppressed below the homogeneous freezing threshold, particularly in the colder upper regions, resulting in an almost complete absence of homogeneous ice nucleation. Under dynamic conditions allowing supersaturations to reach the homogeneous freezing threshold, such as in the ADJ scenario, this prior depletion indirectly enabled the occurrence of homogeneous freezing in later stages.

**Answer to the new comment** Thank you for suggesting an alternative wording. We agree that this part of the text was challenging to convey clearly. We have adopted your suggested version, as it aligns well with the intended message of the manuscript.

**New comments**

Comment 1: Figures 9 and 13: I preferred the previous versions of the Figures, particularly since the model scenarios (STND and AGED / ADJ and HOM) were listed above the two columns. Now, you have to figure the scenarios out from the caption. Please reinsert the abbreviations of the scenarios to the right of the rows.

**Response:** Abbreviations added to the rows accordingly both in Figs 9 and 13.

Comment 2: Figure 10: In the Figure caption it is mentioned

' ... (b) ice saturation profiles from the beginning to the end of the simulation using the STND setup.' I think that the ADJ scenario is meant?

**Response:** This Figure shows the time evolution of the dust and humidity profiles in a STND run. The ADJ uses this end state of STND as a reference initial state. We replaced the end of the sentence to be more clear: ' in a STND run', instead of 'using the STND setup'.

**Comment 3**: Line 377: 'Also the homogeneous freezing is known to be mostly insensitive to the concentration or size distribution of available aerosols, . . . '

Homogeneous freezing is sensitive to the size distribution of the aerosols present, in particular at cold temperatures and higher updrafts. as shown in Baumgartner et al. (2023). Baumgartner, M., Rolf, C., Grooß, J.-U., Schneider, J., Schorr, T., Möhler, O., Spichtinger, P., and Krämer, M.: New investigations on homogeneous ice nucleation: the effects of water activity and water saturation formulations, Atmos. Chem. Phys., 22, 65–91, https://doi.org/10.5194/acp-22-65-2022, 2022.

Response: The text has been modified to also reflect the recent findings on the sensitivity of homogeneous freezing to the size distribution by following form: Homogeneous freezing has often been described as largely insensitive to the concentration or size distribution of available aerosols (Kärcher and Lohmann, 2002; Jensen et al., 2010). However, more recent studies (Baumgartner et al., 2022) indicate that under certain conditions, particularly at very cold temperatures and high updraft velocities, the aerosol size distribution can influence freezing. This choice is justified because the resulting number of solution droplets available for homogeneous freezing remains substantially higher than the concentration of heterogeneous INPs, such that homogeneous freezing still produces higher ice concentrations. Our analysis therefore focuses on how heterogeneous ice nucleation shapes the evolution

of  $S_i$ , while also comparing the relative contributions from homogeneous versus heterogeneous ice nucleation.

Comment 4: Line 555f (of manuscript with changes tracked): 'This reflects the known sensitivity of homogeneous freezing to the magnitude of vertical velocity.' I suggest to add here 'and also the weakening of homogeneous nucleation events due to prior heterogeneous ice nucleation (Spichtinger and Cziczo, 2010).

**Response:** The text was edited according to the suggestion.